# Stochastic and deterministic dynamics of intrinsically irregular firing in cortical inhibitory interneurons

**Philipe RF Mendonça[1], Mariana Vargas-Caballero[2,3], Ferenc Erdélyi[4], Gábor Szabó[4], Ole Paulsen[1], Hugh PC Robinson[1]***

[1]Department of Physiology, Development and Neuroscience, University of Cambridge, Cambridge, United Kingdom; [2]Institute for Life Sciences, University of Southampton, Southampton, United Kingdom; [3]Centre for Biological Sciences, University of Southampton, Southampton, United Kingdom; [4]Division of Medical Gene Technology, Institute of Experimental Medicine, Budapest, Hungary

**Abstract** Most cortical neurons fire regularly when excited by a constant stimulus. In contrast, irregular-spiking (IS) interneurons are remarkable for the intrinsic variability of their spike timing, which can synchronize amongst IS cells via specific gap junctions. Here, we have studied the biophysical mechanisms of this irregular spiking in mice, and how IS cells fire in the context of synchronous network oscillations. Using patch-clamp recordings, artificial dynamic conductance injection, pharmacological analysis and computational modeling, we show that spike time irregularity is generated by a nonlinear dynamical interaction of voltage-dependent sodium and fast-inactivating potassium channels just below spike threshold, amplifying channel noise. This *active irregularity* may help IS cells synchronize with each other at gamma range frequencies, while resisting synchronization to lower input frequencies.

***For correspondence:** hpcr@cam.ac.uk

## Introduction

From the Hodgkin and Huxley model onwards, we have a good understanding of the dynamical basis of regular or periodic firing, and of various kinds of burst firing (*FitzHugh, 1961*; *Hindmarsh and Rose, 1984*; *Hodgkin and Huxley, 1952*). In contrast, the nature of intrinsically irregular firing has resisted elucidation, and appears to be a more complex phenomenon. Irregularity of firing in neurons can arise because of fluctuating patterns of synaptic input due to spontaneous activity (*Destexhe et al., 2001*), or from stochastic fluctuations in the release of transmitter (*Ribrault et al., 2011*). In some regions of the brain, though, certain types of neuron show strikingly high irregularity of firing even when isolated in vitro (*Cauli et al., 1997*; *Grace and Bunney, 1984*; *Ascoli et al., 2008*). The cellular mechanisms of such intrinsic irregularity are unknown, though the stochastic gating of the ion channels involved in spike generation seems likely to play a part. Effective chaos in the nonlinear dynamics of the voltage-dependent ion channels involved in spike generation could also contribute to irregular patterns of membrane potential (*Durstewitz and Gabriel, 2007*; *Fan and Chay, 1994*).

In the cerebral cortex, the function of intrinsically irregular firing is of particular interest. Within the neural circuitry of the neocortex are various types of inhibitory interneuron, several of which have been implicated in the generation of distinct synchronous oscillations at various frequencies from slow (<1 Hz) to very fast (>100 Hz), such as the theta (4–10 Hz), beta (10–30 Hz) and gamma (30–80 Hz) oscillations (Buzsáki, *2006*). For example, the fast-spiking (FS, parvalbumin-expressing, basket morphology) cell network has a crucial role in the emergence of the gamma rhythm

**eLife digest** Neurons send information to other neurons in the brain by generating fast electrical pulses called action potentials (or spikes). When stimulated by input signals of a constant size, neurons generally respond with regular patterns of spiking leading to rhythmical brain activity. However, neurons known as irregular spiking interneurons are unique: the relationship between the input they receive and whether or not they produce a spike appears to be random. The molecular mechanism behind this phenomenon is not clear.

Mendonça et al. set out to investigate whether irregular spiking is truly random, or whether there is some degree of predictability. The experiments used genetically modified mice in which irregular spiking interneurons were specifically labeled with a fluorescent protein, which made them easier to find to record their electrical activity. Sophisticated statistical analyses showed that these neurons are not firing at random. Instead, there is a pattern to the timings of the spikes they produce.

It was previously known that electrical spikes in neurons are generated by sodium ions and potassium ions moving across the membrane that surrounds each cell. Proteins called ion channels provide routes for these ions to pass through the membrane. Mendonça et al. show that compared to other neurons, irregular spiking interneurons have larger numbers of a specific type of potassium ion channel. Mimicking the effect of increasing the number of these potassium ion channels in the interneurons made the firing pattern of these neurons more irregular, while decreasing the number of these channels made the firing patterns more regular and predictable.

A computer model of an irregular spiking interneuron showed that the activity of these potassium ion channels and a type of sodium ion channel plays a key role in producing irregular electrical spiking. Further analysis showed that irregular spiking interneurons can synchronize their activity with fast, but not slow, rhythms in brain activity.

The findings of Mendonça et al. suggest that irregular spiking interneurons can disrupt slow regular electrical activity in the brain. Rhythms in brain activity vary depending on whether we are awake or asleep, and are altered in diseases such as epilepsy and schizophrenia. Now that we have a better understanding of how irregular spiking interneurons work, it should be possible to find out how they coordinate their activity with each other, and what effect they have on animal behavior.

(*Cardin et al., 2009*; *Hasenstaub et al., 2005*). Recent evidence suggests the possibility of a similar specific role for the low-threshold-spiking (LTS, somatostatin-positive, Martinotti) cell network in lower frequency theta or beta rhythms (*Fanselow et al., 2008*; *Vierling-Claassen et al., 2010*). One type of interneuron, however, is distinguished by its intrinsically irregular repetitive firing, showing a broad, apparently random dispersion of its interspike intervals, as opposed to bursting, even when pharmacologically disconnected from any synaptic input. These irregular-spiking (IS) neurons (*Cauli et al., 1997*) seem to have both a distinctive mechanism of spike timing control, and possibly a unique role during synchronous network oscillations.

To enable specific targeting of IS cells, we used a mouse line with green fluorescent protein (GFP) linked to the promoter for *Gad2* (GAD65; *López-Bendito et al., 2004*), in which fluorescently labeled neurons in somatosensory cortex predominantly have an IS phenotype (*Galarreta et al., 2004*). These cells express CCK, VIP and 5HT3a receptors (*Sugino et al., 2006*). They are concentrated in layer 2 (*López-Bendito et al., 2004*), and derive primarily from the caudal ganglionic eminence during development (*López-Bendito et al., 2004*; *Lee et al., 2010*). They connect specifically to each other by gap junctions and mutually inhibitory synaptic connections, which together enable precisely-synchronized irregular firing (*Galarreta et al., 2004*). Their wide axonal arborizations through many layers of the cortex and inhibition of pyramidal cells (*Galarreta et al., 2004*, *2008*) suggest that they could exert a powerful influence on the network. Another distinctive property of these cells is their expression of CB1 cannabinoid receptors, which can suppress their inhibitory output to pyramidal cells, following depolarization of the postsynaptic cell (*Galarreta et al., 2008*). Although they make up a large proportion of inhibitory interneurons in superficial layers, they have received much less attention than other classes of interneuron, such as FS and LTS cells.

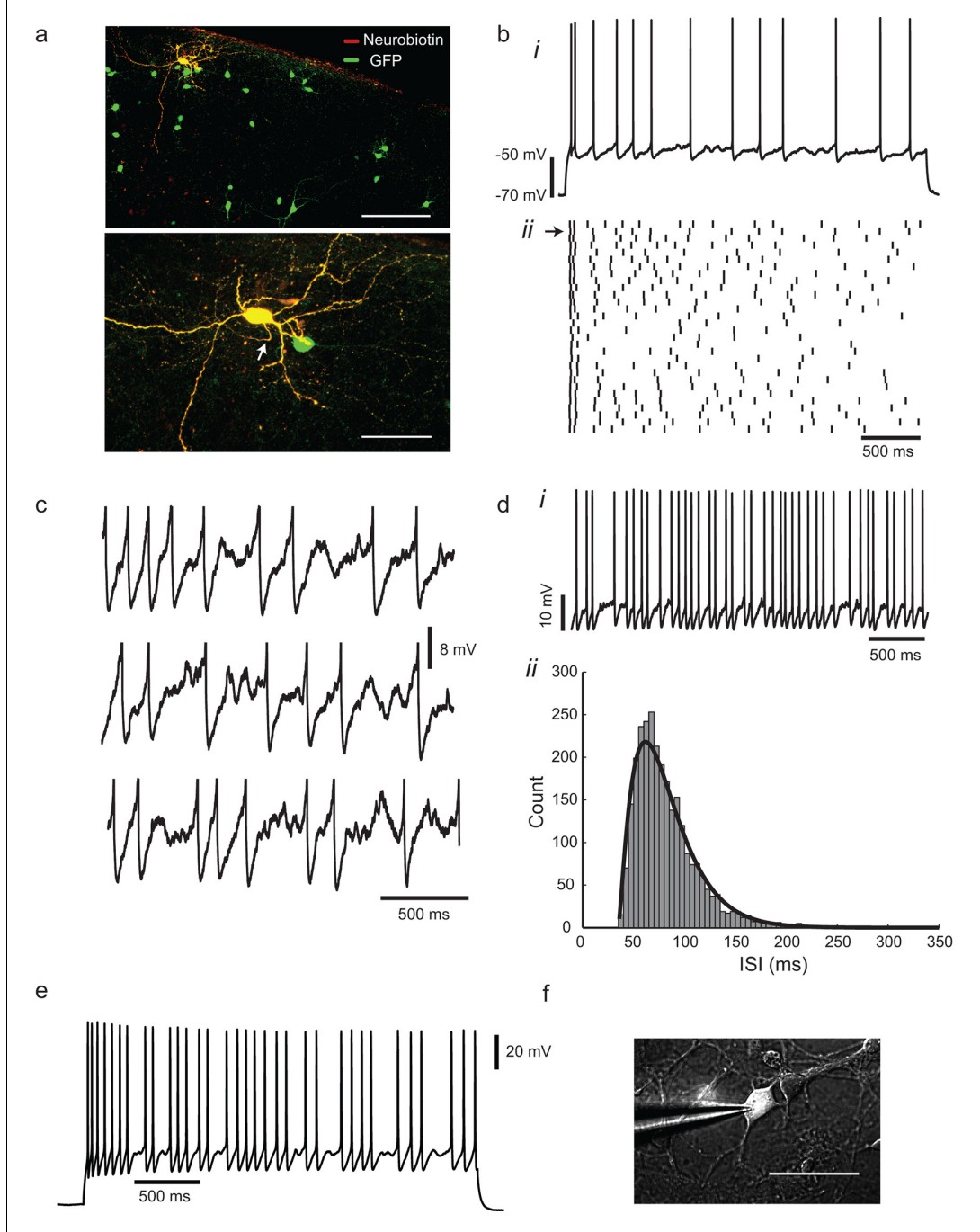

**Figure 1.** Irregular-spiking in a population of cortical inhibitory interneurons. (**a**) Distribution of *Gad2*-GFP mouse neurons in the somatosensory cortex (top). Below is the detailed morphology of a typical irregular-spiking interneuron which was filled with neurobiotin. White arrow indicates the axon initial segment. Irregular-spiking *Gad2*-GFP interneurons were consistently found in superficial layers and displayed noticeably bigger somata. Stacked confocal images of cells in a 300 µm thick slice; scale bars, 150 µm and 50 µm. (**b**), (**i**) Irregular spiking in response to a constant 120 pA current step. Resting potential was −68 mV. After an initial fast spike doublet, firing settles into an irregular pattern of spikes, separated by noisy fluctuations of membrane potential. (**ii**) Raster plot of spike times in 30 successive responses to the same current step, separated by 10 s intervals. Spike train corresponding to (**bi**) is indicated by an arrow at left. (**c**) Close-up view of the interspike membrane potential fluctuations in three consecutive trials from the ensemble shown in (**bii**). Spikes have been truncated. (**d**), (**i**) Higher frequency firing in another cell, excited by a 220 pA constant current stimulus. (**ii**) The distribution of 2730 interspike intervals (ISIs) in one cell, fitted with a gamma distribution: $f(t) = \frac{1}{\Gamma(n)\tau}\left(\frac{t-t_r}{\tau}\right)^{n-1}\exp\left(\frac{t_r-t}{\tau}\right), \quad t > t_r$ where *n* is 2.29, τ is 20.7 ms, and refractory period $t_r$ is 35.05 ms. CV(ISI) = 0.38, mean firing frequency is 13.6 Hz. (**e**), *Gad2*-GFP cortical interneurons display the same irregular-

*Figure 1 continued on next page*

*Figure 1 continued*

spiking pattern in primary culture (12–16 DIV; n = 10). Irregularity increased with development, and was observable even at higher firing frequencies (15–20 Hz) as in cortical slices. (f) Patch-clamp recording of a GFP+ neuron in culture. Scale bar 50 μm.

In this study, we ask: what mechanisms underlie the striking irregularity of firing, and what are the functional consequences of this in an oscillating cortical network? Using a combination of patch-clamp recording in slices of somatosensory cortex, time series analysis and computational modeling, we show that IS neurons generate robust, intrinsically irregular firing by nonlinear interactions of voltage-dependent currents and channel noise. The degree of irregularity is tuned by the level of a fast-inactivating potassium conductance, and voltage-dependent sodium and potassium channel openings contribute a high level of voltage noise at threshold. The effect of these mechanisms is that these cells reject synchronization to a low frequency (10 Hz), while synchronizing effectively to higher, gamma frequencies, a property which could give them a prominent role in gating local cortical gamma oscillations.

## Results

### A genetically-defined population of irregular-spiking cortical interneurons

In the cortex of *Gad2*-GFP mice, fluorescent cell bodies are concentrated in layer 2, with dendrites concentrated in layers 1 and 2/3 and axons which ramify through the cortical layers (*Figure 1a*). The morphology of fluorescent neurons was varied, with bitufted, bipolar and multipolar cells observed, as described by *Galarreta et al. (2004)*. Cells had input resistances of 331 ± 164 MΩ and passive time constants of 15.4 ± 7.7 ms (mean ± SD, n = 82). In response to a step current stimulus in a whole-cell current-clamp recording, 77% (82/106) of the cells showed a characteristic pattern of action potentials (APs) at irregular intervals, with fairly deep and slow afterhyperpolarizations, often following an initial adaptation phase (*Figure 1bi*). Irregular spiking interneurons displayed larger somata ($\approx$15 μm diameter) and more prominent projections than did the remaining 23% of GFP+ neurons, which had a regular-spiking response (excluded from analysis, except when stated), as described by *Galarreta et al. (2004, 2008)*.

The irregular trajectory of action potential intervals in IS neurons varied from trial to trial (*Figure 1bii*), and the membrane potential showed quite large, variable fluctuations between spikes (*Figure 1c*). Over long periods of continuous stimulation, the distribution of interspike intervals was skewed and unimodal, and could be reasonably well-fitted by a gamma function (*Figure 1d*). Irregularity was quantified as the coefficient of variation of interspike intervals or CV(ISI), the ratio of the standard deviation of intervals to their mean (see Materials and methods), which is equal to 1 for a Poisson point process, and 0 for a perfectly periodic process. CV(ISI) was reduced at higher stimulus levels and firing frequencies, and was quite variable from cell to cell, but ranged from 0.1 (fairly regular) to 0.6 at a firing frequency of $\approx$10 Hz ($CV_{10\ Hz}$ = 0.28 ± 0.15, n = 45). The irregularity persisted in the presence of blockers of ionotropic glutamate and $GABA_A$ receptors and is therefore presumably generated intrinsically, rather than by noisy synaptic input. The intrinsic nature of the IS was confirmed in primary cultures of dissociated *Gad2* neurons, which displayed a similar spiking pattern, despite simpler morphology and reduced connectivity (*Figure 1e,f*).

### Recurrence of sequences of irregular interspike intervals

To characterize the dynamics of irregular spiking, we first examined return maps of interspike intervals – scatter plots of each interval against its predecessor – which displayed no discernible fine structure (*Figure 2a,b*). We therefore looked at the predictability of higher-order sequences of intervals using recurrence plots (*Eckmann et al., 1987*; *Marwan et al., 2007*). First, sequences of interspike intervals were embedded – that is, translated into all sub-sequences of length m, the embedding dimension – each of which defines a point in m-dimensional embedding space, and can be thought of as a piece of 'recent history'. For example, *Figure 2c* (top) illustrates two similar embedding points of dimension m = 3 occurring within two different interval sequences. Similarity

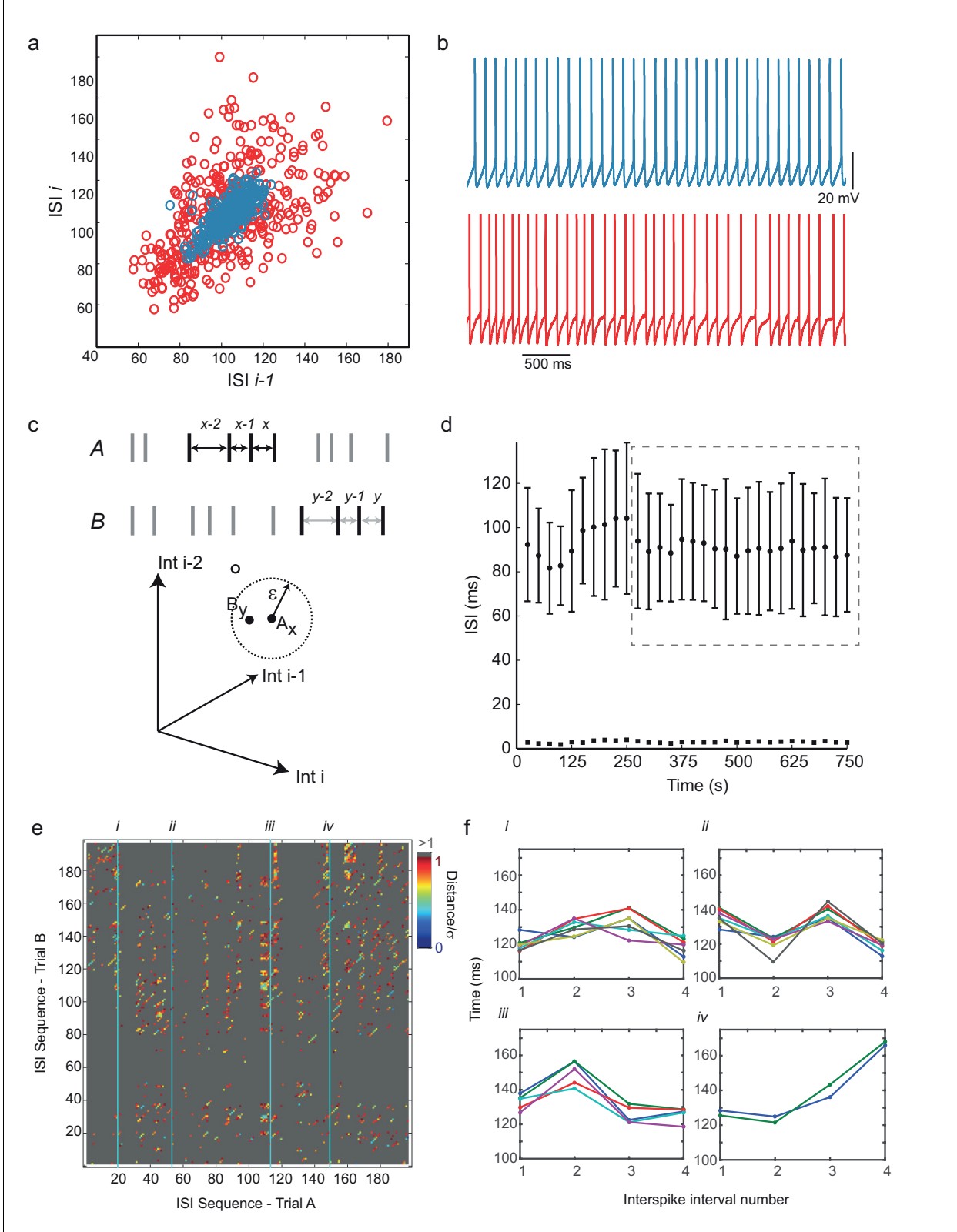

**Figure 2.** Predictability and nonlinearity of interspike interval sequences. (a) Examples of two contrasting ISI return maps extracted from a regular-spiking cell (blue, mean frequency 9.67 Hz, CVISI = 0.075) and an irregular-spiking cell (red, mean frequency 9.65 Hz, CVISI = 0.207). (b) Segments of corresponding spike trains. (c) Principle of recurrence analysis. The dynamical state of the process is represented by vectors of consecutive ISIs, or embedding points. In this example, point $A_x$ in a 3-dimensional embedding of an interspike interval sequence A (whose coordinates are ISIs$x$-2, $x$-1 and

*Figure 2 continued on next page*

*Figure 2 continued*

x) is similar to point $B_y$ in interspike interval series B (top), because their distance is less than a threshold ε (bottom). (d) Selection of stationary sequences of stimulus trials for recurrence analysis. The mean ISI in each trial lasting 8 s, repeated at 25 s intervals, is plotted with its standard deviation (filled circles and error bars), and the standard error of the mean (filled squares). Sections of the time series were accepted as sufficiently stationary if the average trial-to-trial change in mean ISI was less than half the average standard error of the mean (e.g. region shown in dashed gray rectangle). (e) Example cross-recurrence plot between two consecutive stimulus trials, A and B, embedding dimension m = 4, ε = one standard deviation of the ISIs. Position (x,y) is colored according to the Euclidean distance between the length-4 ISI sequences at position *x* in A and *y* in B. Thus blue points reflect recurrence of very similar patterns. (f) Four examples of repeated patterns or 'motifs' of ISIs in sequence B corresponding to the patterns at positions (i)–(iv) in sequence A, as indicated in (e). See *Figure 2—figure supplement 1* and *Figure 2—source data 2* for recurrence plot quantification.

The following source data and figure supplement are available for figure 2:

**Source data 1.** Numerical values for *Figure 2d*.
**Source data 2.** Table showing details of recurrence plot analysis in ten cells.
**Figure supplement 1.** Significance testing of recurrence and determinism of interspike interval sequences.

of dynamical state is measured by proximity in this space (*Figure 2c*, bottom), and this can be generalized to any *m*. A cross-recurrence plot of two sequences of intervals, A and B, for example two successive spiking responses to an identical step current stimulus, is a matrix in which element (*x,y*) has a value representing the distance between the *x*th embedding point of A and the *y*th embedding point of B (*Eckmann et al., 1987*; *Marwan et al., 2007*) (see Materials and methods for further details). *Figure 2e* shows an example in which Euclidean distance in embedding space is represented by color, so that close recurrences show up as colored dots on a gray background. Diagonal lines of slope one, many examples of which can be seen in *Figure 2e*, indicate periods when the trajectory of one time series evolves similarly to the trajectory of the other. Examples of four different recurrent ISI sequence 'motifs', identified from the recurrence plot in *Figure 2e*, are shown in *Figure 2f*. Recurrence can be quantified as follows. Applying a threshold to the cross-recurrence plot, so that element (x,y) = 1 if the distance between $A_x$ and $B_y$ is less than a threshold neighborhood size ε (see *Figure 2d*), or zero otherwise, gives a binary cross-recurrence plot, in which the density of 1's is defined as the degree of recurrence, and the fraction of these which lie within diagonals of length 2 or greater is defined as the degree of determinism. Randomly shuffling the time series before embedding destroys significant recurrence and determinism (allowing statistical testing of their significance (see *Figure 2—figure supplement 1*, and Materials and methods). We calculated cross-recurrence plots between successive pairs of trials (5–30 s in duration), omitting the first 450 ms of firing in each trial to exclude initial adaptation, for 10 neurons which showed long periods of stationary responses (*Figure 2d*), at average firing frequencies between 4 and 17 Hz. Using a standard sequence size or embedding dimension *m* = 4, and a neighborhood size (ε) of one standard deviation of the ISIs, we found that in five of ten cells, both recurrence and determinism were significant ($p < 0.05$, z-test), while only recurrence was significant in a further two cells, and in the three remaining cells, neither recurrence nor determinism were significant. See *Figure 2—source data 2* for details. Note that, unlike the related technique of nonlinear prediction (*Kantz and Schreiber, 1997*; *Sprott, 2003*), the detection of significant recurrence and dynamical determinism by recurrence plot quantification is less confounded by nonstationarity, and relatively insensitive to the exact choice of *m* and ε. Thus, irregular sequences of spikes generated during a constant stimulus in about half of IS neurons show recurrent, correlated sequences of four or more successive intervals. It seems likely that the concerted action of voltage-dependent ion channel populations would be involved in producing such determinism. We found similar recurrence and determinism in a conductance-based biophysical model of these cells, described below, when applying the same analysis procedure to its spike trains (Figure 6, *Figure 6—figure supplement 1*). We noted that those cells that failed to show significant recurrence and determinism had particularly strong voltage noise in their interspike intervals (not shown).

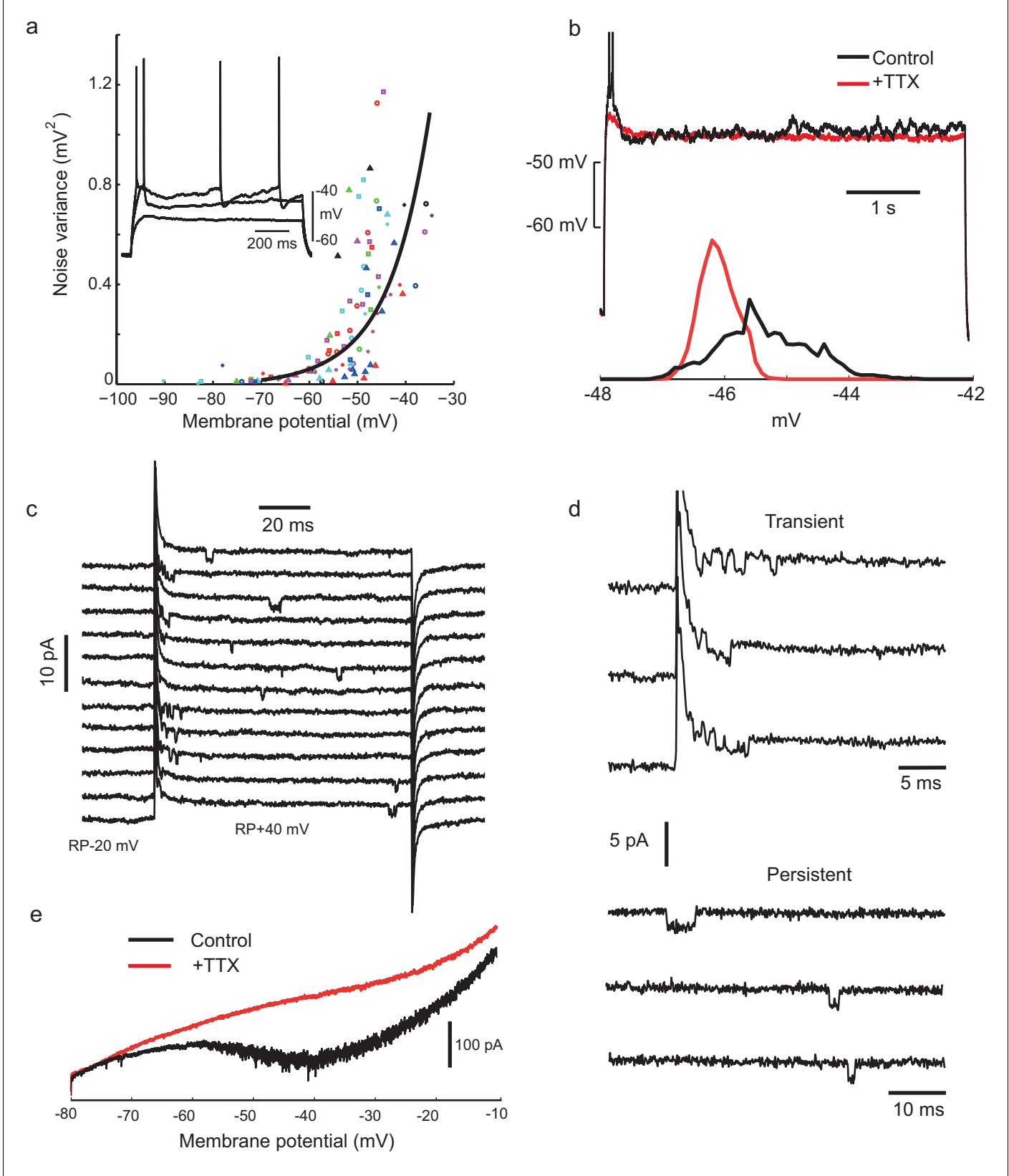

**Figure 3.** Voltage-gated sodium channel activation is required for noisy subthreshold voltage fluctuations. (**a**) The amplitude of subthreshold fluctuations (see example waveform in inset) rises sharply above a threshold membrane potential (≈ −50 mV). Measurements for 23 cells indicated by

*Figure 3 continued on next page*

*Figure 3 continued*

different symbols. The curve shows a fit to a model of combined NaP (1950 channels) and $g_{Kt}$ (180 channels) single channel noise, see Materials and methods for details). Inset: three example traces for one cell during step current stimulation of 60, 90 and 100 pA, showing the onset of membrane potential noise. (b) Fluctuations are blocked by applying tetrodotoxin (TTX, 100 nM). Membrane potential traces in another IS cell with and without perfusion of TTX, in response to the same current step, which is subthreshold in the steady-state after an initial doublet (top). Corresponding amplitude histogram of the membrane potential (bottom). (c) Membrane current in a cell-attached patch in response to repeated depolarizing steps, from RP-20 mV to RP+40 mV, as indicated. RP = resting membrane potential. Sodium channel openings are both transient, within 10 ms of the depolarization, and persistent, occurring late in the depolarization. (d) Transient and persistent openings at higher time resolution. (e) Whole-cell recordings confirming the presence of a TTX-sensitive, non-inactivating inward current at the firing threshold potential range (−55 mV). A slowly depolarising ramp (20 mV/s, from −80 mV to −10 mV, −70 mV holding potential) was applied in the presence of TEA (2 mM), 4-AP (2 mM) and $Cd^{+2}$ (200 μM) in order to eliminate $K^+$ and $Ca^{+2}$ currents, with TTX (500 nM) added during the trial shown in red.

## Voltage-dependent sodium channel openings are required for voltage fluctuations

Next, we investigated the biophysical mechanisms which underlie the irregular firing. Clearly, one potentially relevant phenomenon is the noisy fluctuation in the membrane potential which switches on above −50 mV (*Figure 3a*). We found that these fluctuations depended on voltage-gated sodium channels, since they were eliminated by applying tetrodotoxin (TTX; *Figure 3b*, n = 6 cells). To further investigate the unitary properties of voltage-gated sodium channels, we carried out cell-attached recordings in somatic patches. Characteristic ≈20 pS inward openings were observed, concentrated soon after the beginning of the depolarization (*Figure 3c*), with an extrapolated reversal potential of about +120 mV positive to the resting potential, as expected for single voltage-gated sodium channels (*Sigworth and Neher, 1980*). We also observed frequent late openings of the same channel amplitude, up to 100 ms following +40 mV depolarizations from rest, (*Figure 3c and d*, in 4 out of 5 patches containing transient Na channels). Whole-cell recordings further confirmed the presence of a non-inactivating, TTX sensitive inward current, evoked in response to a slowly depolarizing ramp (*Figure 3e*, prominent in 11/13 cells), when $K^+$ and $Ca^{2+}$ currents were reduced with TEA (2 mM), 4-AP (2 mM) and $Cd^{2+}$ (200 μM). Similar 'persistent' sodium current (NaP) and channel openings have been described in many neurons and excitable cells (*Kiss, 2008*). Thus, stochastic, voltage-dependent gating of sodium channels could be involved in generating irregularity of firing. Sodium-channel-driven subthreshold noise has been observed in other cell types (*White et al., 1998*), but without appearing to produce the high level of firing irregularity observed in IS cells at ≈10 Hz firing frequencies (*Alonso and Klink, 1993*). The deterministic recurrence of the interspike intervals suggests that another active mechanism might also be involved.

## A fast-inactivating potassium current activates around threshold

Both voltage-gated and calcium-activated potassium channels contribute to spike repolarization and spike after hyperpolarizations in cortical neurons. However, neither blockers of calcium-activated potassium channels (iberiotoxin and apamin) nor intracellular perfusion of a fast calcium buffer (BAPTA) diminished irregularity of firing (see *Figure 4—figure supplement 1*), and we concluded that intracellular calcium signaling is not centrally involved in the dynamics of intrinsic irregularity. We therefore next examined the voltage-dependent potassium currents, which are of key importance in determining action potential generation and shape (*Bean, 2007*). In particular, we focussed on those whose voltage-dependence of gating might allow dynamical interaction with the sodium channels.

Whole-cell voltage-clamp of the outward currents in response to families of step depolarizations revealed an early transient outward or A-type potassium current (*Figure 4a*), which could be isolated by applying a pre-pulse protocol (*Amarillo et al., 2008*, *Maffie et al., 2013*) in the presence of 5 mM TEA to remove slower $K^+$ currents (*Figure 4b*, n = 9). Fits of the voltage-dependence of the peak conductance and of the steady-state inactivation of this transient potassium conductance ($g_{Kt}$) showed that activation and inactivation curves overlapped around the threshold (*Figure 4c*, n = 18 cells for inactivation, n = 36 cells for activation), peaking within 1–2 ms, and inactivating over about 20–30 ms (*Figure 4d*, top). Additionally, this fast inactivating outward current recovered from inactivation with a time constant of around 40 ms at −70 mV (*Figure 4d*, bottom). These properties are

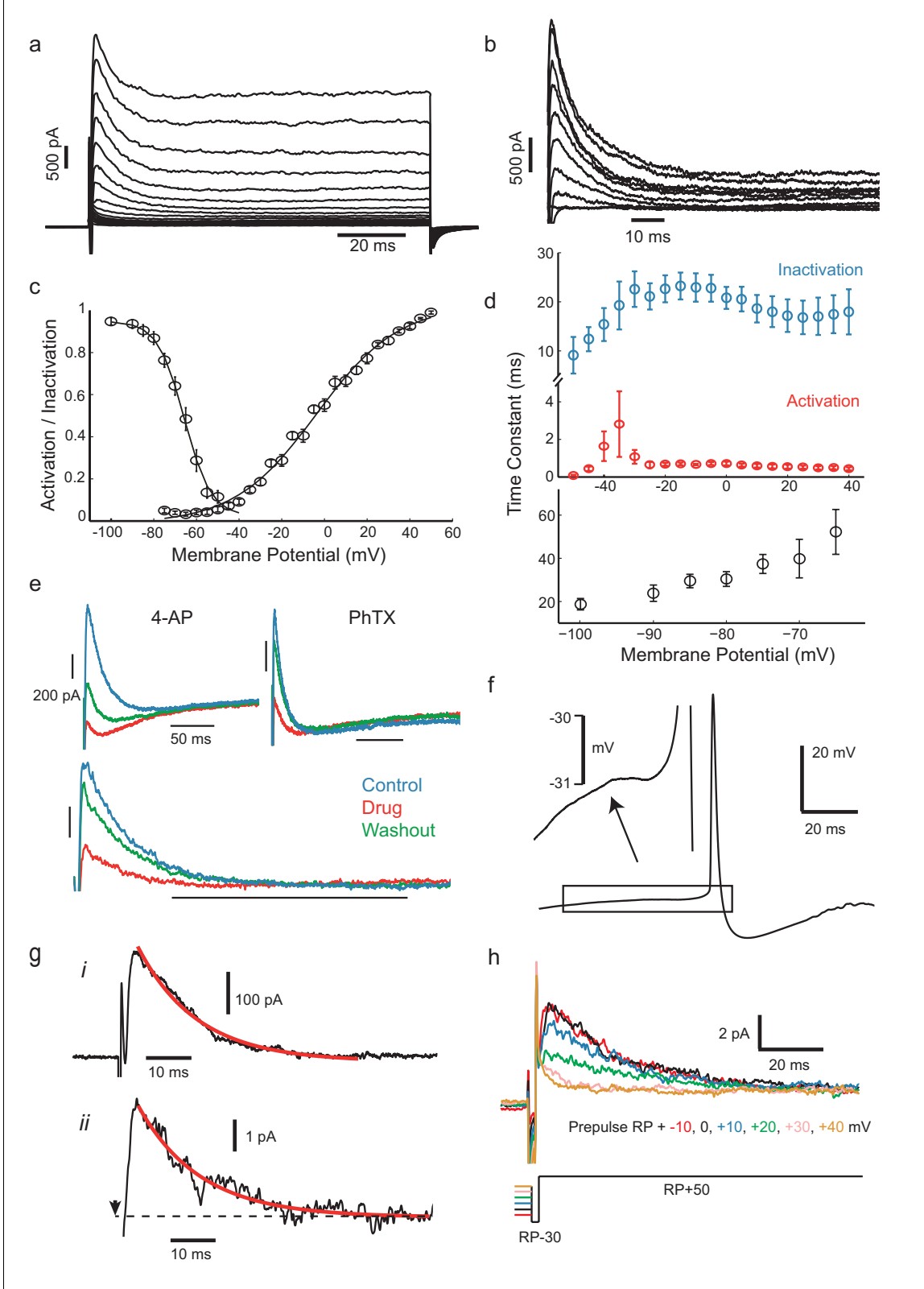

**Figure 4.** IS neurons express a fast transient outward current with similar kinetics to Kv4. (**a**) Whole-cell currents in response to a family of voltage steps from −80 to 0 mV in 5 mV steps. (**b**) A-type current separated from other outward current components. The remaining step-evoked current following a pre-pulse (−30 mV, 200 ms) capable of inactivating A-type current was subtracted from total current. Voltage steps from −50 to +40 in 10 mV steps. Recordings were carried out in the presence of 5 mM TEA in order to block slowly activating K⁺ currents. (**c**) Voltage-dependence of steady-state

*Figure 4 continued on next page*

*Figure 4 continued*

activation and inactivation. (**d**) Activation (red) and inactivation (blue) time constants of dissected A-type current (top) and recovery of inactivation time constant (bottom). (**e**) The fast inactivating outward current found in these cells was sensitive to the A-type current blocker 4-AP (7 mM) and the Kv4-specific blocker phrixotoxin (PhTX, 5 μM). Top panels: total currents in control, drug application, and washout, as indicated. Lower panel: PhTX block of transient outward current fraction, separated as in (**b**). (**f**) Average of 537 aligned APs following ISIs lasting longer than 100 ms shows a small prespike dip or inflection, attributed to the activation of the transient $K^+$ current. (**g**), (**i**) Isolated transient outward current with a single exponential fitted to the decay phase ($\tau$ = 13.05 ms). (**ii**) Example current from a cluster of transient $K^+$ channels in a cell-attached patch (step from RP-30 mV to RP+50 mV at the time indicated by arrow, outward current plotted upwards), fitted with the same exponential time constant as in (**i**). (**h**) Dependence of patch current on the potential of a 500 ms prepulse before a step from RP-30 to RP+50 mV, showing that it inactivates over the range RP+10 mV to RP+40 mV.

The following source data and figure supplements are available for figure 4:

**Source data 1.** Numerical values for *Figure 4c,d and e*.

**Figure supplement 1.** Irregularity is not diminished by buffering intracellular calcium.

**Figure supplement 2.** Fast inactivating outward current is insensitive to TEA (2 mM).

**Figure supplement 3.** *Gad2*-GFP cortical interneurons from primary cultures display the two conductances required for spiking irregularity.

not consistent with Kv1 channels (the current was insensitive to 1 μM α-dendrotoxin, n = 4, not shown), including Kv1.4 (recovery from inactivation in the range of milliseconds rather than seconds, see *Wickenden et al., 1999*), nor with channels from the Kv3 family (transient currents were TEA insensitive, see *Figure 4—figure supplement 2*). The gating properties closely resemble those of Kv4-family voltage-dependent potassium channels in pyramidal neurons (*Birnbaum et al., 2004*), and this was further supported by its sensitivity to 4-AP (*Figure 4e* top, n = 6) and the specific Kv4.2/4.3 blocker phrixotoxin (PhTX; *Figure 4e* bottom, n = 7), which produced a partial, reversible block of 55% at a concentration of 5 μM. We fitted conventional Hodgkin-Huxley type models to the voltage-step responses of this current (see Materials and methods), and estimated a peak transient conductance at 0 mV ($g_{max0}$, see Materials and methods) of 22.37 ± 14.41 nS (n = 8 cells, mean ± SD). This current would be expected to delay the rise of membrane potential just before spike initiation. Although the membrane potential leading into spikes was generally highly fluctuating, averaging the waveform of hundreds of action potential, aligned with the fastest point of the upstroke, consistently showed the presence of a clear dip or inflection in the rising phase, about 10 ms before the start of the fast upstroke (*Figure 4f*), which we attribute to this current. There was also a high density of single channel currents in some cell-attached patches (n = 7) with similar activation and inactivation properties (*Figure 4g and h*), implying some clustering in the membrane, as previously described for Kv4 channels (*Alonso and Widmer, 1997*; *Jinno et al., 2005*). The fast and small-amplitude single channel openings in these recordings were not well-resolved, but appeared to comprise step transitions corresponding to a single channel chord conductance of about 10–12 pS (assuming $E_K \approx -90$ mV). The single channel conductance of Kv4 channels is not extensively-characterized, but reports vary from ≈5 pS to ≈20 pS in low potassium external solutions, and it is sensitive both to external potassium concentration and to association with accessory proteins such as KChIPs (*Holmquist et al., 2002*; *Cooper and Shrier, 1989*). Thus, overall, the transient potassium conductance recorded at the soma strongly resembles reported descriptions of Kv4-mediated conductance.

## Transient outward conductance determines spike irregularity

To test whether and how this inactivating $K^+$ current is involved in generating irregular firing, we injected a synthetic dynamic conductance (*Robinson and Kawai, 1993*; *Sharp et al., 1993*) with the kinetics and voltage-dependence measured from voltage clamp, which should have the same electrical effect as the native conductance at the soma. Artificial conductance injection of $g_{Kt}$ was sufficient to modulate the spiking irregularity of intrinsically irregular *Gad2* interneurons (*Figure 5a–c*, *Figure 5—figure supplement 2*). When negative $g_{Kt}$ was injected, i.e. *subtracting* from the dynamics of the native conductance in these cells (as shown in *Figure 5d* for voltage clamp currents), we saw a

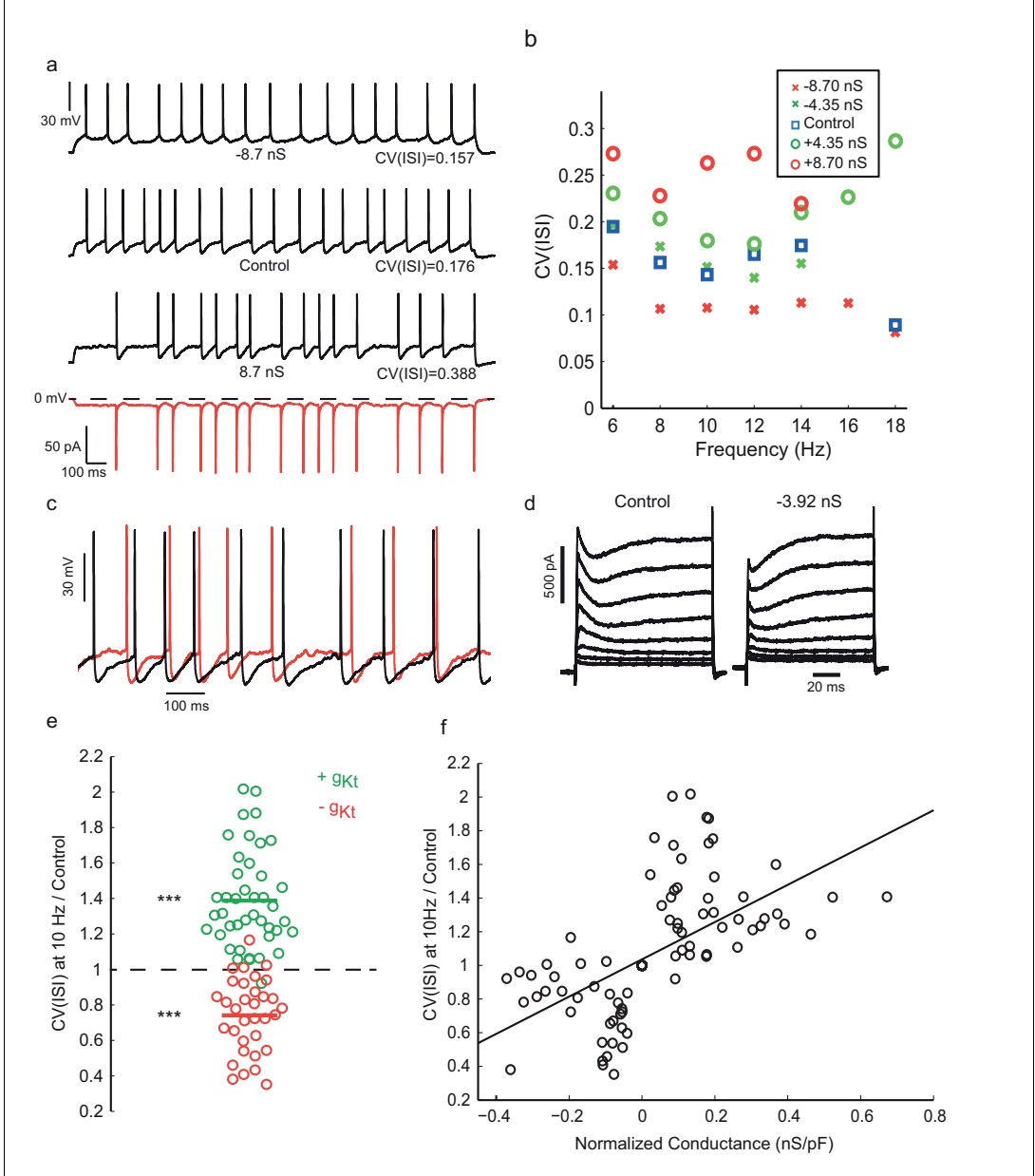

**Figure 5.** Injection of synthetic $g_{Kt}$ modulates spiking irregularity. (a) Positive and negative $g_{Kt}$ injection in the same cell at the same frequency range (8–10 Hz). While −8.7 nS injection ($g_{max0}$, See Materials and methods) caused a reduction in the AHP amplitude and regularized the firing pattern, injecting +8.7 nS created more evident noisy plateaus before some APs, resulting in more irregular firing. Red bottom trace shows the current passed during the positive $g_{Kt}$ conductance injection (outward, hyperpolarizing current plotted downwards). (b) Effect of $g_{Kt}$ on spiking irregularity in another cell, showing its consistency over different firing frequencies. (c) Close-up of the membrane potential trajectories from (a), +$g_{Kt}$ (red) superimposed on control (black), showing extended and increased noisy subthreshold fluctuations produced by the $g_{Kt}$ conductance. (d) Potassium currents during a family of step depolarizations from −80 mV to −60, −50, … +10 mV. Subtraction of 3.92 nS of the fast-inactivating Kv current by dynamic-clamp largely cancels the transient component, leaving a residual, non-inactivating delayed rectifier current. (e) Relative changes in CV(ISI) at 10 Hz firing frequency induced by addition or subtraction of $g_{Kt}$ conductance. Data from 42 cells: points are individual measurements, with some cells measured at two or more different conductance levels. Wilcoxon non parametric test, $p<9.8 \times 10^{-16}$ for positive $g_{Kt}$, and $p<1.6 \times 10^{-8}$ for negative $g_{Kt}$. (f) Relationship between relative change in CV(ISI) at 10 Hz firing frequency and injected $g_{Kt}$ conductance. The linear regression fit is superimposed. Pearson's correlation r = 0.59, $p<2.66 \times 10^{-12}$.

The following source data and figure supplements are available for figure 5:

**Source data 1.** Numerical values for *Figure 5e and f*.

*Figure 5 continued*

**Figure supplement 1.** Injecting a shunting conductance at the soma, causing a large reduction in input resistance, modifies the action potential amplitude and shape, and divides down membrane potential fluctuations, but does not regularize firing.

**Figure supplement 2.** Example spike patterns for three different cells with (**a**) negative $g_{Kt}$ conductance injection and three different cells with (**b**) positive $g_{Kt}$ injection, showing decreased and increased irregularity respectively.

**Figure supplement 3.** Effect of pharmacological block of A-type current in IS cells is consistent with the effect of the negative $g_{Kt}$ injection.

striking regularization of firing in the range of frequency examined, as well as a reduction in the afterhyperpolarization (AHP) amplitude. On the other hand, injecting positive $g_{Kt}$ induced an increase in the irregularity of firing, accompanied by a more prominent subthreshold membrane potential fluctuation between spikes (*Figure 5c*). The effect of $g_{Kt}$ on CV(ISI) was consistent especially at lower firing frequencies (e.g. 10 Hz, *Figure 5e*, n = 42 cells), and it was even more evident when the total $g_{Kt}$ injected was normalized to the capacitance of each cell, which is proportional to its plasma membrane area (*Figure 5f*). Pharmacological block of $g_{Kt}$ by 4-AP or phrixotoxin gave a similar result to negative conductance injection, reducing irregularity of firing (See *Figure 5—figure supplement 3*).

## Mechanisms of firing variability in a simple model of IS neurons

Having shown experimentally that the transient potassium current plays a key role in controlling irregular firing in IS neurons, we sought to understand how it might do so, by studying a computational model of these cells. We constructed a conductance-based biophysical model, in which the key $g_{Kt}$ and NaP conductances could be modeled either as stochastic or deterministic elements. A two-compartment model was used, comprising a somatic compartment which contained voltage-dependent conductances, linked to a passive dendritic compartment. The dendritic compartment was included in order to capture, in a simplified way, the extended spatial aspect of the cell morphology. Similar to a widely-used model of fast-spiking inhibitory interneurons (*Erisir et al., 1999*; *Gouwens et al., 2010*), the soma included Kv1 and Kv3 voltage-dependent potassium conductances and a sodium conductance. To this, however, was added a $g_{Kt}$ conductance based on the voltage-clamp findings above, and a persistent sodium conductance (NaP). NaP and $g_{Kt}$ were modeled either deterministically or stochastically with a dynamic noise variance (see Materials and methods for details).

In the deterministic model, interspike intervals were of two types: long, almost stationary pauses, and periods of subthreshold oscillation, of unstable and variable amplitude, at a frequency of about 28 Hz (*Figure 6a*). In a three-dimensional subspace of the (8-dimensional) phase space of the model, displaying the activation variable of $g_{Kt}$ as $x$, the membrane potential as $y$, and the sodium inactivation variable as $z$, some of the dynamical structure underlying this behaviour can be seen (*Figure 6b*, *Video 1*). The subthreshold oscillations correspond to variable numbers of circuits around an unstable-amplitude cycle in one region of phase space, before the system escapes into the upstroke of a spike. Long pauses correspond to a transition to another critically slow region of phase space where $h$, $m$, and $V$ remain at an almost fixed point, while Kv1 activation ($n$) slowly subsides, eventually leading to an escape from this region, either directly into a spike, or into a period of subthreshold oscillations. Thus, this set of conductances gives two dynamical mechanisms for generating irregular interspike intervals: variable numbers of circuits of unstable amplitude subthreshold oscillations, and long pauses in a slow region of phase space. The activation of $g_{Kt}$ is seen to vary considerably for different spikes (note the spread in values of $m_{Kt}$ in the afterhyperpolarization in *Figure 6b*).

Changing $g_{Kt}$ and NaP conductances to a stochastic form somewhat obscures the difference between the pauses and subthreshold oscillations, causing more irregular and variable fluctuations in the subthreshold oscillations (*Figure 6c*). The subthreshold noise amplitude is highly dependent on the stochastic $g_{Kt}$, since it is greatly reduced if $g_{Kt}$ is deterministic (*Figure 6d*). It is only slightly reduced if NaP is deterministic, but greatly reduced if all voltage-gated sodium conductance is removed (*Figure 6d*, 'TTX' - right hand side). This implies that subthreshold fluctuations are dominated by $g_{Kt}$-driven stochastic fluctuations which are strongly amplified by the voltage-gated sodium

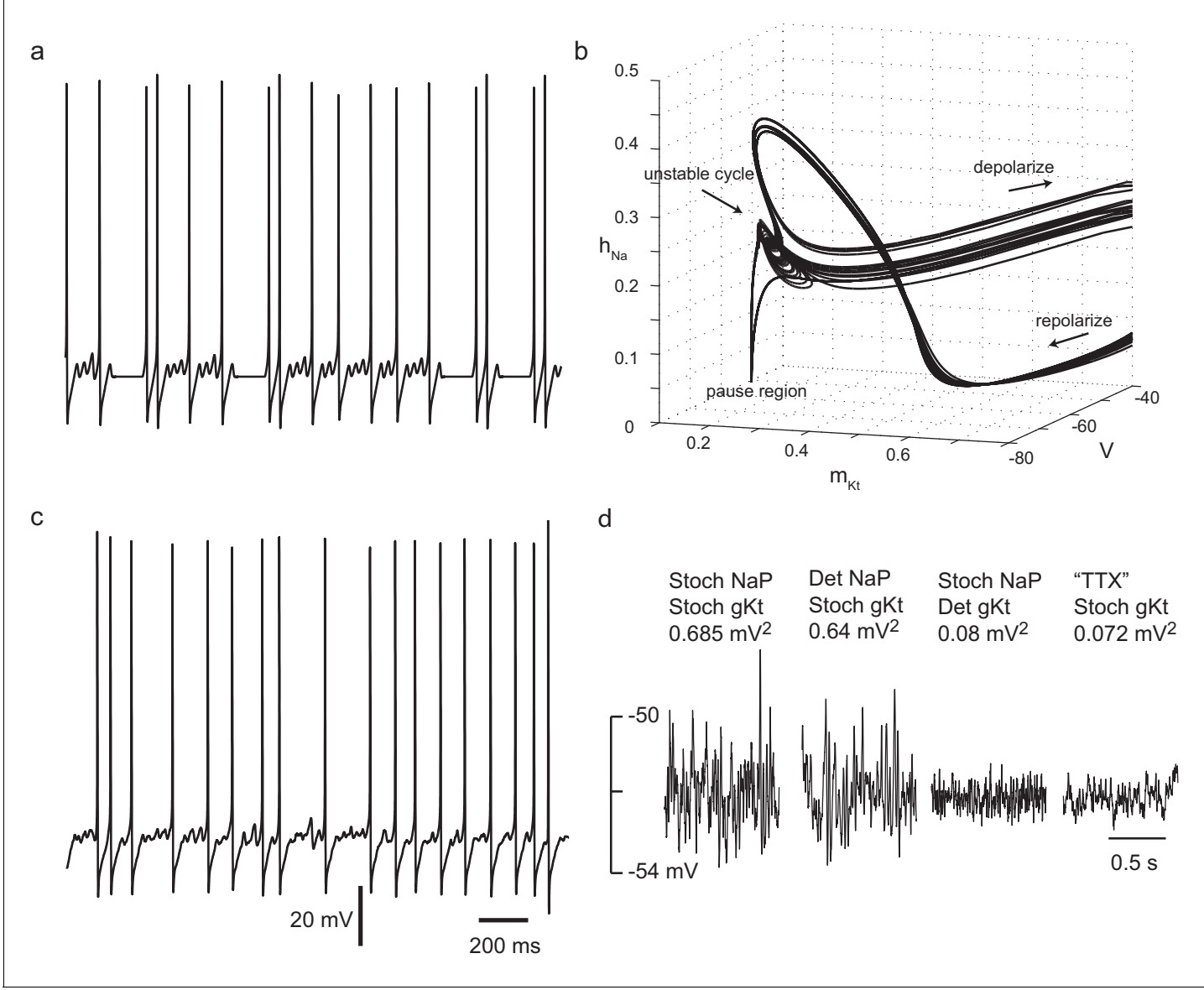

**Figure 6.** Irregular firing in a simple biophysically-based model. (**a**) Two-compartment model with Nav, Kv1, Kv3 and $g_{Kt}$-type conductances shows complex spike timing, as a result of unstable subthreshold oscillations and trapping in a nearly-fixed state. $\overline{g}_{kt}$ = 7 nS, stimulus current, 100 pA. For other parameters, see Materials and methods. (**b**) Unstable subthreshold oscillations and a fixed-point 'ghost' seen in the phase trajectory of the model with zero noise in the ($m_{Kt}$, $h_{Na}$, V) subspace (101 pA, 7 nS $\overline{g}_{kt}$). (**c**) Adding noisy non-inactivating (persistent) sodium channel conductance(equivalent to 500 channels) and noisy $g_{Kt}$ (equivalent to 7 nS or 700 channels) masks subthreshold oscillations, but preserves high spike irregularity. Stimulus current 90 pA. (**d**) $g_{Kt}$ channel noise is strongly amplified by voltage-dependent sodium conductance. Subthreshold membrane potential noise for a stimulus current of 72 pA, with 7 nS $\overline{g}_{kt}$ and 10 nS $\overline{g}_{NaP}$, either stochastic or deterministic, and for the case in which all sodium current is blocked ('TTX'), and stimulus current of 90 pA, to polarize the membrane to the same range of membrane potential as without sodium current.

The following figure supplement is available for figure 6:

**Figure supplement 1.** Statistics and significant recurrence and determinism of time series generated by the computational model.

conductance – both are required. The greater importance of $g_{Kt}$ noise over NaP noise is largely due to its much longer correlation time (10 ms versus 1 ms), which means that $g_{Kt}$ noise is much less filtered by the membrane time constant. Thus, these strong subthreshold membrane potential fluctuations appear to be actively-amplified channel noise, somewhat like noise-driven subthreshold

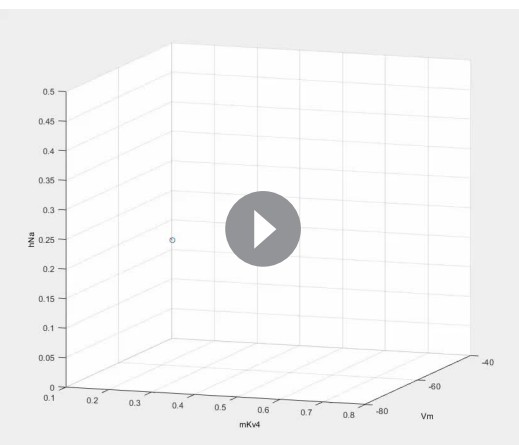

**Video 1.** Movie showing dynamics in phase space of the deterministic model. Corresponds to the trajectory shown in *Figure 6b*.

oscillations, as described in entorhinal stellate neurons (*Dorval and White, 2005*). This result suggests that, although the fit of the subthreshold membrane potential noise variance by the 'voltage-clamped' channel noise of NaP and $g_{Kt}$ channels (*Figure 3a*, see Materials and methods) appears to describe the onset of this noise reasonably well, the numbers of channels are probably overestimated, as the powerful active amplification of fluctuations is not taken into account.

The action of $g_{Kt}$ in promoting irregular firing across the range of frequencies is visualized in *Figure 7*, in which the firing frequency is plotted as a function of both stimulus current and the amount of $g_{Kt}$ conductance included in the model, with the surface colored to indicate the CV(ISI). As the amount of $g_{Kt}$ in the membrane is increased to 12 nS, a region of structurally-stable variability is created for firing frequencies up to 20–30 Hz, above which frequency the CV(ISI) subsides, as seen in recordings, in both deterministic (*Figure 7a*) and stochastic (*Figure 7b*) forms of the model. In the deterministic form of the model, CV(ISI) reaches values of ≈1, much higher than experimentally observed (red region of surface in *Figure 7a*). However, the addition of noise dilutes the irregularity of this high-CV region to ≈0.3 (*Figure 7b*), as seen experimentally. The stochastic model shows a much more linear firing frequency-current (*f-I*) characteristic, as observed in actual recordings – i.e. the dynamic noise linearizes the input-output relation of these neurons. The distribution of ISIs produced by the stochastic model also resembles experimental distributions (*Figure 6—figure supplement 1*, panel a).

## Synchronization of irregular firing through gap-junctions in networks of model IS neurons

Irregular spiking could exert a far greater impact in the cortical network if it were synchronized amongst IS neurons, which are connected with each other in a specific gap-junction-coupled network (*Galarreta et al., 2004*). However, the mechanism of irregularity proposed here depends on the intrinsic dynamics and noise sources within individual cells. It seems possible that the impact of fluctuations generated within individual cells could be diluted when cells are connected in an electrical network. Therefore, we simulated small networks of symmetrically-coupled stochastic IS neurons. In a 5-cell network, firing became highly synchronous as gap junction conductance was increased, as seen in the sharp central peak of cross-correlation (*Figure 7c*). However, CV(ISI) was maintained at the same level as for uncoupled cells, even with strong coupling and complete synchrony (*Figure 7d*). This perhaps non-intuitive result implies that in effect, nonlinearly-amplified fluctuations are cooperative amongst cells and are well-coupled by the current flow through gap junctions.

## Synchronization to oscillating input and the function of IS neurons

The intrinsic irregularity of firing of IS neurons, which is distinctive amongst the cell types of the cortical network, raises the question of what these neurons do, particularly in the context of the regular firing which underlies organized oscillations in many frequency bands (*Buszáki and Draguhn, 2004*). This particular type of IS neuron directly inhibits pyramidal neurons, and it has been suggested that it might promote asynchronous firing and thereby resist synchronous oscillations (*Galarreta et al., 2008*). In order to test how these cells integrate periodic inputs, we examined their ability to synchronize their spikes to rhythmic oscillation in a naturalistic stimulus consisting of several conductance components: a stationary, noisy AMPA receptor-type excitatory conductance and an oscillating (10 Hz or 40 Hz) GABA$_A$ receptor-type shunting inhibitory conductance, combined with simultaneously adding or subtracting $g_{Kt}$ using dynamic-clamp (*Figure 8a*). *Figure 8b* shows an

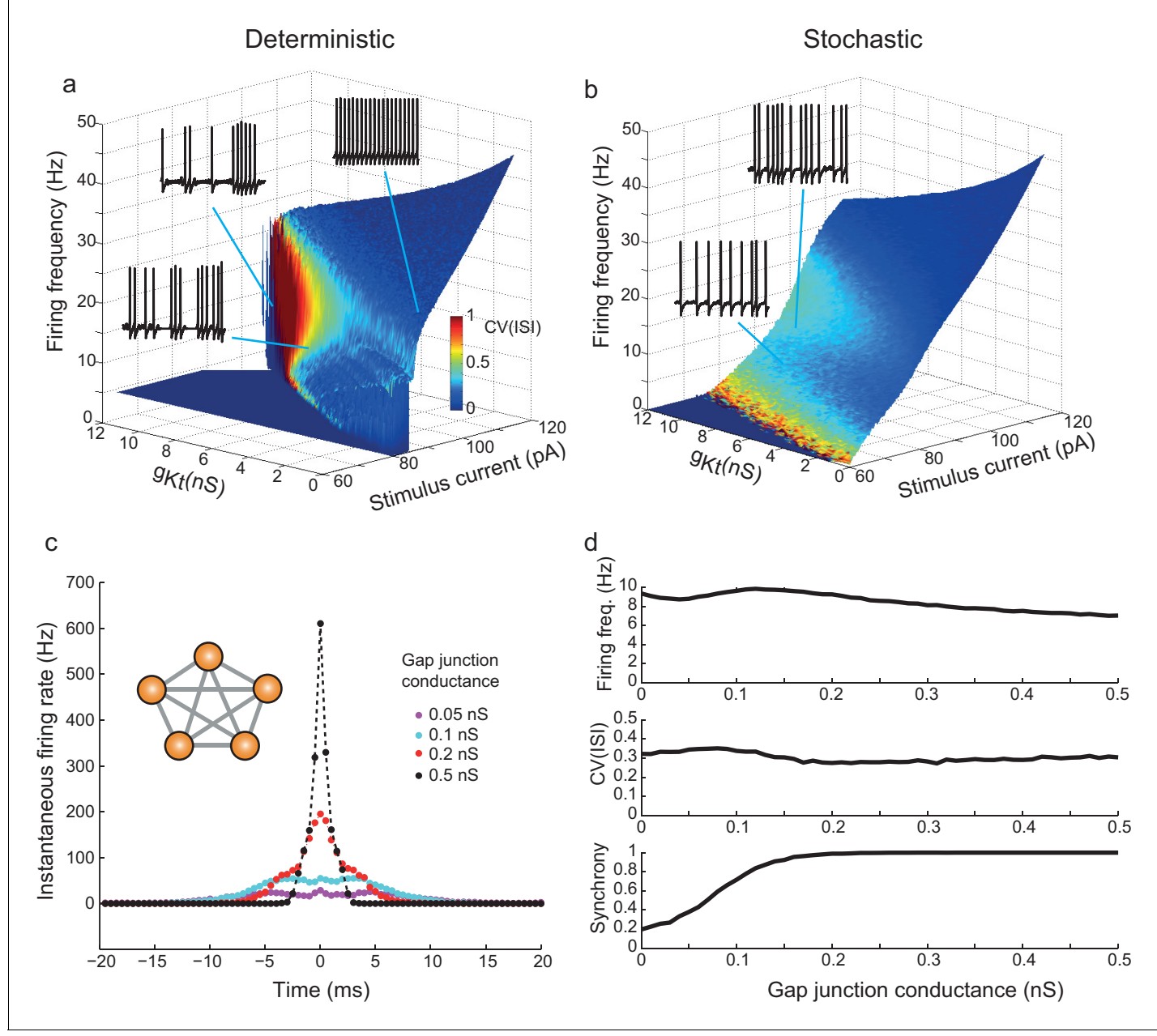

**Figure 7.** $g_{Kt}$ enhances irregularity in deterministic and stochastic biophysical models. (**a**) Surface showing the dependence of firing frequency on the total $g_{Kt}$ and stimulus current level, colored according to the CV(ISI) of firing. Regions of low CV(ISI) correspond to periodic firing, while regions of high variability arise through the pausing and unstable subthreshold oscillation mechanisms. (**b**) Analogous plot for the stochastic model containing voltage-dependent noise fractions due to 1000 persistent sodium channels, and different numbers of 10 pS $g_{Kt}$ channels equivalent to the conductance indicated. Inset example voltage traces (1 s of firing): (**a**) bottom: 101 pA, $\bar{g}_{Kt}$ 7 nS; top left: 106.7 pA, 10 nS $\bar{g}_{Kt}$; top right: 89 pA, $\bar{g}_{Kt}$ 0.5 nS. (**b**) bottom: 90 pA, 500 $g_{Kt}$ channels (= 5 nS); top: 95 pA, 500 $g_{Kt}$ channels. (**c,d**) Irregularity in simulated gap-junction-coupled ensemble of IS cells (700 $g_{Kt}$ channels (= 7 nS), 500 NaP channels (= 5 nS)). (**c**) Cross-correlation of spike trains in one pair of neurons within a symmetrically-connected network of five IS neurons (inset), each excited by a constant stimulus of 90 pA. Exact synchrony appears as coupling is strengthened, as indicated by the single sharp peak centered on 0 ms. See Materials and methods, Spike Analysis, for details of calculation of cross-correlation. (**d**) Firing frequency, CV(ISI) and synchrony – the fraction of spikes in one cell which occur within ± 10 ms of spikes in the other cell – as a function of the gap-junctional conductance. CV(ISI) is undiminished even for highly synchronous firing, with strong gap-junctional conductance.

example of an IS cell subjected to an elevation of $g_{Kt}$ (+3.57 nS $g_{max0}$). This depresses the synchrony of spikes to the $g_{GABA}$ rhythm (*Figure 8bii,iii*) across the range of oscillation amplitudes tested. Conversely, subtraction of $g_{Kt}$ from another cell (*Figure 8c*) enhanced synchrony over a wide range of oscillation amplitudes. These striking effects of $g_{Kt}$ on synchrony to 10 Hz inputs are not observed, however, for 40 Hz input (*Figure 8d*, summary statistics for the whole set of cells at both frequencies are shown in *Figure 8e*). Thus modulation of irregularity of IS neurons by the level of $g_{Kt}$ (see *Figures 5* and *6*) appears to determine their ability to synchronize to oscillatory inhibition, and the dynamics of $g_{Kt}$ are such that it can counteract 10 Hz but not 40 Hz rhythms. Rejection of synchronization results from the intrinsically irregular dynamics at lower frequencies, around 10 Hz, while resonance with a noise-obscured subthreshold oscillation (whose frequency in the deterministic model is 28 Hz) could contribute to the stronger synchronization at higher frequencies. Thus the native $g_{Kt}$ of IS cells allows them to resist synchrony to lower network frequencies such as 10 Hz, while complying readily with higher, gamma frequency rhythms. This could have the effect of destabilizing lower frequency network oscillations while helping to stabilize higher-frequency rhythms, and help to determine the times of onset and offset of organized gamma-frequency firing in the network.

## Discussion

Here we have used a combination of experiment and modeling to show that the voltage-dependent gating and stochastic activation of fast-inactivating potassium and sodium channels play major roles in generating the intrinsic irregularity of cortical irregularly-spiking (IS) inhibitory interneurons. We also showed that at frequencies matching firing frequencies where this irregularity is high (up to 20 Hz), these cells strongly reject synchronization to the naturalistic oscillating input. This finding is especially relevant considering that irregular-spiking VIP interneurons fire at 10–15 Hz in vivo (whisking and non-whisking activities, *Lee et al., 2013*).

IS cells have been hard to define functionally, because of the profusion of types of inhibitory interneuron in the cortex, and because irregular-spiking behavior may also arise from fluctuations in synaptic input or membrane integrity during recordings. The development of a genetically-modified mouse in which intrinsically IS cells are labeled with GFP has allowed targeted study of a relatively homogeneous population of IS neurons (*Galarreta et al., 2004*, *2008*; *López-Bendito et al., 2004*). Inducible in vivo genetic fate mapping (*Miyoshi and Fishell, 2011*; *Miyoshi et al., 2010*) has been used to show that these IS interneurons originate from the caudal ganglionic eminence relatively late in development (E16), express 5HT3a receptors, VIP and calretinin, and form about 10% of CGE-derived interneurons, which dominate the more superficial layers of cortex and comprise about 30% of all cortical interneurons. Within upper layer 2, the lamina in which they are concentrated, IS cells may make up a large proportion, perhaps 50% (*López-Bendito et al., 2004*), of interneurons. Though we know quite a lot about their functional synaptic connectivity, and its regulation by CB1 receptors (*Galarreta et al., 2004*, *2008*), the origin of the irregular spiking behavior itself has remained unknown.

Predictability of spike trains of irregular-spiking cortical neurons has been examined in a previous study (*Englitz et al., 2008*), which concluded that the variability is not a consequence of low-dimensional, effective deterministic processes. However, that study did not examine the genetically-defined population of neurons studied here. In contrast, we found that there was both significant recurrence and determinism, i.e. predictability, in sequences of spike intervals, in about half of the cells examined (*Figure 2*). We propose that this predictability is linked to the dynamics of a prominent low-threshold fast-inactivating voltage-gated potassium conductance interacting with voltage-dependent sodium conductance, including a persistent fraction, which enhances the activation of sodium channels around AP threshold (*Figures 3* and *4*). Evidence for this was the sensitivity of membrane fluctuations to TTX, the strong modulation of irregularity by injecting artificial inactivating $K^+$ conductance, and the ability to reproduce this phenomenon in a biophysical model (*Figures 6* and *7*).

We suggest that the fast-inactivating $K^+$ current that we found in these cells is likely to be mediated by Kv4 potassium channel subunits, for several reasons. Not only does its fast recovery from inactivation ($\approx$40 ms) exclude the other main candidate, Kv1.4, but the current was partially blocked by PhTX, and had a weak voltage dependence for activation and inactivation kinetics, which are known properties of Kv4-mediated currents. IS cells in this same GFP mouse model have been

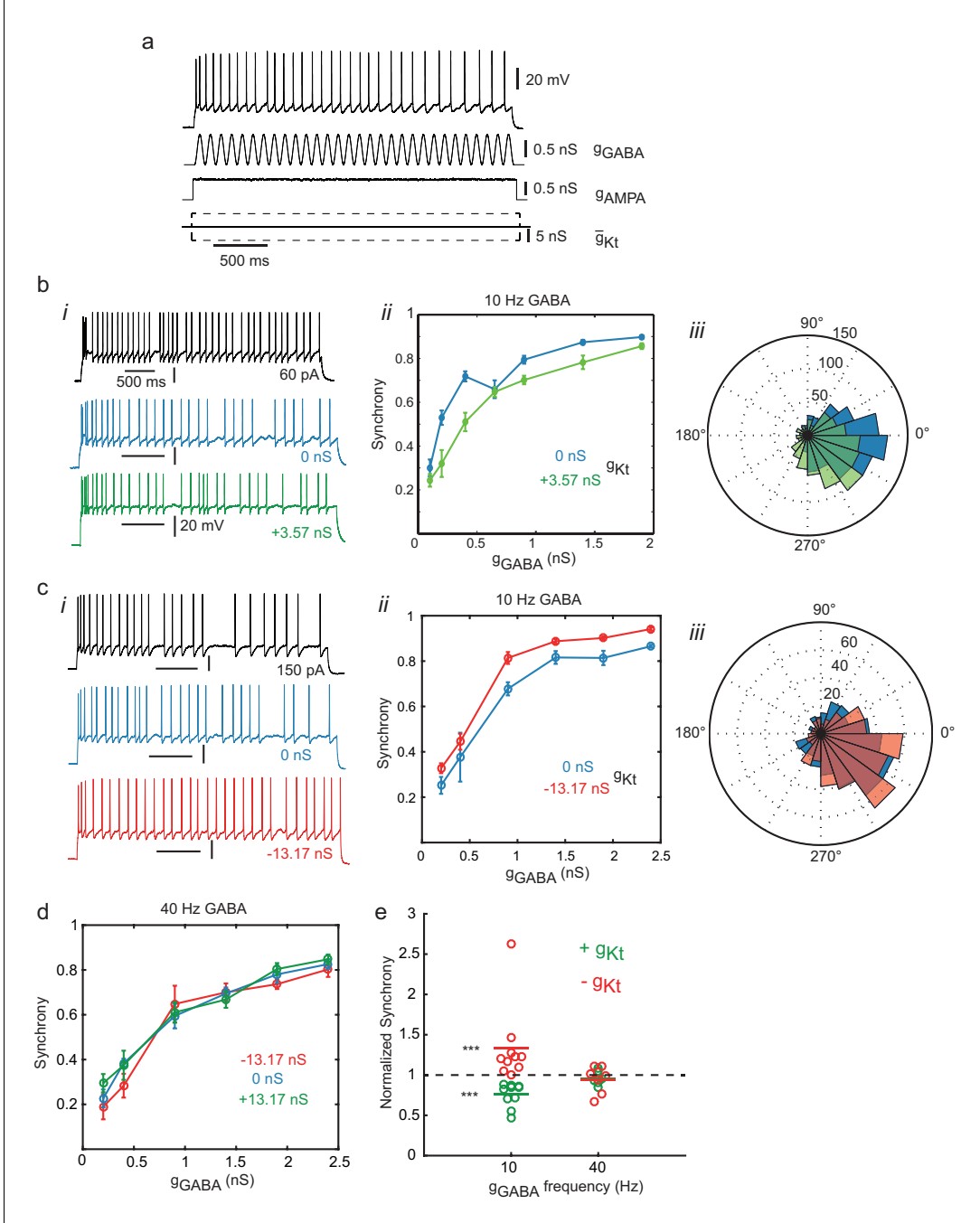

**Figure 8.** Synchronization to oscillating inhibition is controlled by $g_{Kt}$. (a) Naturalistic stimulus protocol. The cell was stimulated with a constant step of AMPA conductance ($g_{AMPA}$, reversing at 0 mV) with added conductance Ornstein-Uhlenbeck noise (standard deviation 2% of the step amplitude, $\tau = 5$ ms), combined with a sinusoidal GABA$_A$ conductance ($g_{GABA}$, reversing at −60 mV) and introduction of positive, zero or negative $g_{Kt}$. $g_{AMPA}$ was adjusted so that the cell fired close to the frequency of the $g_{GABA}$ inhibitory oscillation. (b) Effect of adding $g_{Kt}$ on a slightly irregular-firing cell at 10 Hz $g_{GABA}$. (i) Step current response (black), response to oscillatory conductance stimulus with (green) or without (blue) addition of 3.57 nS $g_{Kt}$. (ii) Spike entrainment synchrony (see Materials and methods, Spike Analysis) to the 10 Hz $g_{GABA}$ oscillation as a function of the oscillation amplitude. Synchrony rises progressively with oscillation amplitude in control (blue), and is depressed by the addition of 3.57 nS $g_{Kt}$ (green). (iii) Spike phase histogram for pooled responses to lower amplitude $g_{GABA}$ oscillations (up to 1 nS), showing a reduction in the sharpness of synchrony. (c) Example of subtracting $g_{Kt}$ in another irregular-firing cell at 10 Hz $g_{GABA}$. (i) example responses. (ii) Subtraction of $g_{Kt}$ (red) increased synchrony to $g_{GABA}$ oscillation over a wide range of amplitudes, when compared to control (blue). (iii) spike phase histograms for pooled responses up to 1 nS $g_{GABA}$ oscillations. Subtraction of $g_{Kt}$ enhances the phase preference. (d) Lack of effect of $g_{Kt}$ on synchronization to 40 Hz (gamma) oscillation. (e) Summary of effects of $g_{Kt}$ perturbation on synchrony in different cells. Each symbol denotes an experiment on an individual cell, showing the ratio of synchrony, evaluated at ≈ 1/3 of the

*Figure 8 continued on next page*

*Figure 8 continued*

maximum $g_{GABA}$ amplitude applied in each case, during $g_{Kt}$ injection, normalized to its control value with no injection. 10 Hz: $g_{Kt}$ addition (n = 10, green) and subtraction, (n = 10, red); 40 Hz: $g_{Kt}$ addition (n = 6, green) or subtraction (n = 7, red). At 10 Hz, but not 40 Hz, $g_{Kt}$ perturbation has a significant effect. Wilcoxon nonparametric rank sum test, p=6.5 × 10$^{-5}$ for both positive and negative $g_{Kt}$, and p=0.36 and 0.69 at 40 Hz for positive and negative $g_{Kt}$ respectively.

The following source data is available for figure 8:

**Source data 1.** Numerical values for *Figure 8b,c,d and e*.

shown to express high levels of Kv4.2 (kcnd2) mRNA, higher than in a population of pyramidal cells and seven times higher than in a population of fast-spiking interneurons (*Sugino et al., 2006*), but only very low levels of Kv1.4 (kcna4) transcripts. The kinetics and voltage dependence of activation and inactivation also match well those described for Kv4.1/4.2 in pyramidal neurons (*Birnbaum et al., 2004*). However, further work will be needed to prove definitively the identity of these channels.

Interestingly, we found that GFP+ cells in dissociated primary cortical cultures also showed robust intrinsic irregular firing (*Figure 1e*), and expressed transient K$^+$ and persistent Na$^+$ currents (*Figure 4—figure supplement 3*). This suggests that normal morphology and circuit formation in development are not required for the irregularity. It is possible that the relatively high mature input resistance of IS neurons (331 MΩ), compared to other types of interneuron could directly lead to greater variability, since single-channel voltage noise should be bigger. However, we found that when we injected an intense (2 nS) static shunting conductance, reversing at −70 mV, near to the resting potential – effectively greatly reducing input resistance, the action potential is reduced in amplitude, and afterhyperpolarizations and interspike membrane potential fluctuations are strongly diminished, but irregularity is not reduced (*Figure 5—figure supplement 1*). Likewise, adding the artificial K$^+$ conductance, which also decreases input resistance, caused increased, not decreased firing variability. This indicates that irregularity is produced by a more complex dynamical mechanism, driven partially by stochastic channel opening, but also dependent on the nonlinearity of the transient K$^+$conductance.

The model that we have implemented suggests two deterministic active mechanisms for high spike time variability: long 'pause' states, in which the dynamical state is presumably trapped near the 'ghost' of a fixed point, and unstable subthreshold oscillations. Both these mechanisms exist over a fairly wide range of values of $g_{Kt}$ conductance density and stimulus level (*Figure 7*), but depend on the presence of $g_{Kt}$. The high irregularity and variability of the purely deterministic form of the model is suggestive of deterministic chaos, although a rigorous proof of chaos in the model would require for example proof of a positive Lyapunov exponent, and is beyond the scope of this study. At the same time, however, it is clear that significant dynamical noise must be involved to some extent in the irregularity, as a result of the single-channel characteristics of the main voltage-dependent channels involved. ISI distributions in IS neurons appear to be shaped by the single-channel current fluctuations around AP threshold (*Figures 3* and *4*), both of sodium channels including persistent ones, and of $g_{Kt}$ channels. Adding noise in the model, which mimics the single-channel activity of these channels around the AP threshold, realistically obscures the regularity of the subthreshold oscillation, and leads to ISI distributions very similar to those observed experimentally (*Figure 6—figure supplement 1*, panel a). While still clearly preserving the $g_{Kt}$-induced region of high-CV firing (*Figure 7b*), the addition of noise changes the unnaturally high CV(ISI) of the deterministic model (≈1) to a value compatible with the experiments (≈0.3). Thus, we believe that the interaction of both elements, the nonlinear deterministic Hodgkin-Huxley equations and the single-channel dynamical noise is needed for an adequate description of irregular spiking. Both the deterministic and stochastic components of the model have measurable biophysical parameters.

Other, related dynamical models of irregular firing in neurons have been proposed. For example, a bifurcation analysis (*Golomb et al., 2007*) of an FS cell model, incorporating different levels of slowly-inactivating potassium current ($I_d$, probably corresponding to Kv1 channels) showed that higher levels of this current can produce 'stuttering' behavior associated with subthreshold oscillations, as also seen experimentally in FS neurons (*Tateno et al., 2004*). Dispersion of interspike

intervals produced by variations in amplitude of subthreshold oscillations of a much higher frequency (100–150 Hz) characterized in spinal motoneurons has been termed 'mixed-mode' oscillation (*Manuel et al., 2009*, *Iglesias et al., 2011*). Noise-induced switching between a fixed point and a spiking limit cycle has been shown to produce high irregularity in the Hodgkin-Huxley model (*Rowat, 2007*). Recently, Stiefel et al. suggested that fast-activating K$^+$ currents could promote this kind of switching behavior in IS neurons, leading to high irregularity (*Stiefel et al., 2013*). Although the mechanism that we propose here is both more specific and more complex, the basic necessity for an interaction between noise and strong nonlinearity assisted by fast K$^+$ channels is consistent with these studies. Interestingly, A-type potassium and NaP currents have also been implicated in the generation of theta-frequency (5–10 Hz) membrane potential oscillations in hippocampal interneurons (*Morin et al., 2010*, *Skinner, 2012*), possibly through the dynamical mechanism of 'critical slowing', in which the amplitude of noise-driven fluctuations grows near a bifurcation. This may also be relevant in IS neuron membrane potential fluctuations, and further studies of their sensitivity to noise near AP threshold would be merited.

The active irregularity produced by coordinated activation of populations of voltage-dependent channels and their activation-dependent single-channel noise, which we propose, may have at least two important advantages. First, it is an energetically-favorable way to generate high spike interval irregularity in individual cells, while minimizing unnecessary membrane potential fluctuation, because fluctuations switch on sharply just below AP threshold, and their active amplification makes them highly effective at controlling spike timing. Second, IS cells are linked to each other in a specific gap-junction-coupled network, and also inhibit each other through GABA$_A$ receptor-mediated synapses (*Galarreta et al., 2004*, *2008*). This would be expected to enhance the local synchrony of irregular firing (*Gouwens et al., 2010*), potentially greatly increasing its impact on network activity. The partly active, deterministic nature of irregularity and the subthreshold dynamics would help to coordinate the sources of irregularity in different cells, via current flow through gap junctions. Examining the synchronization of ensembles of stochastically-modeled IS neurons connected through gap junctions (*Figure 7c,d*), we found that even with high gap-junctional coupling and resultant complete synchrony of firing, irregularity is maintained just as high as in isolated neurons. This non-trivial result implies that the network of IS neurons can indeed fire with both high irregularity and precise synchrony.

Overall, these results suggest that coordinated irregular firing is important for the cortex. Synchronous oscillation, although it is a population activity that is relatively easily detected and studied, and which may provide a timing mechanism for processing, is also low-dimensional and limited in its capacity to represent information. Synchronous irregular firing may help to create diverse network firing patterns, useful in representation of information, and to find solutions to optimization problems in pattern recognition. It may also enhance STDP-based learning (*Christodoulou and Cleanthous, 2011*), and could be important in decision-making and generation of spontaneous choices. The coupled network of IS neurons could also control initiation and termination of periods of synchronous regular oscillations, consistent with the rejection of synchronization to low-frequency rhythms that we observed (*Figure 8*), which was enhanced by the addition of artificial $g_{Kt}$.

In conclusion, we have provided evidence that, in addition to the direct effect of stochastic channel noise, IS neurons have a specific nonlinear deterministic mechanism that drives spike time irregularity. The mechanism depends on a nonlinear interaction of Kv4 potassium and sodium channels around AP threshold. This novel mechanism appears to allow this group of neurons to have a coordinated, and hence powerful, impact on concerted activity in the cortex.

## Materials and methods

### Animals and tissue preparation

A genetically-modified mouse line was used, in which GFP was linked to the promoter for *Gad2* (*López-Bendito et al., 2004*). At ages between P30 and P60, animals were sacrificed in accordance with the UK Home Office regulations under the Animals (Scientific Procedures) Act of 1986, and 300 µm sagittal slices of the neocortex were cut with a tissue slicer (Leica VT1200S, Leica UK, Cambridge), using standard techniques described elsewhere (*Morita et al., 2008*; *Kim and Robinson, 2011*). Slices were observed using an upright microscope (Olympus BX51WI, XLUMPlanFI 20X/

0.95W objective) with infrared illumination and an oblique condenser, combined with epifluorescence to visualize GFP-expressing neurons.

Primary cultures of dissociated cortical *Gad2*-GFP neurons were obtained by methods similar to those described by *Schroeter et al. (2015)*. Extrahippocampal cortex was isolated at E17 or P0, and cultured for 12–17 days in vitro. All protocols followed UK Home Office regulations for the care and use of animals.

## Solutions

During recording, slices were superfused with a solution containing (mM): 125 NaCl, 25 NaHCO$_3$, 2.5 KCl, 1.25 NaH$_2$PO$_4$, 2 CaCl$_2$, 1 MgCl$_2$, 0.01 glycine, 25 D-glucose, maintained at a pH of 7.4 by bubbling with 95% O$_2$, 5% CO$_2$ gas mixture. In most experiments, 10 μM CNQX, 10 μM APV and 10 μM gabazine were added to silence background synaptic activity in the slice. For whole-cell recordings, the following pipette filling solution was used (mM): 105 K gluconate, 30 KCl, 10 4-(2-hydroxyethyl)-1-piperazineethanesulfonic acid (HEPES), 4 ATP-Mg, 0.3 GTP-Na, 10 creatine phosphate-Na (adjusted to pH 7.3 with KOH, −10 mV liquid junction potential (LJP)). In recordings with elevated calcium buffering (see text), the concentration of K gluconate was reduced to 90 mM, and 10 mM 1,2-bis(o-aminophenoxy)ethane-N,N,N',N'-tetraacetic acid (BAPTA)-Na was added. In the case of whole-cell recordings of persistent sodium currents, the following intracellular solution was used (mM): 90 Cs methanesulfonate, 30 CsCl, 10 BAPTA, 10 HEPES (adjusted to pH 7.3 with HCl, −12 mV LJP). For cell-attached recordings, the following pipette solution was used (mM): 150 NaCl, 2.5 KCl, 12.5 tetraethylammonium (TEA) chloride, 2 CaCl$_2$, 1 MgCl$_2$, 10 HEPES. Potassium currents were measured with 300 nM TTX added to the bath solution. Blockers were dissolved in a HEPES-buffered artificial cerebrospinal fluid and puff-applied through a glass pipette of around 50 μm in tip diameter. Salts were obtained from Sigma-Aldrich (Dorset, UK), and channel and receptor blockers from Tocris Bioscience (Bristol, UK), with the exception of phrixotoxin, which was acquired from Abcam (Cambridge, UK). Recordings were carried out at 30–33 °C.

## Electrical recording

Whole-cell recording in current-clamp and voltage-clamp modes, and cell-attached single-channel recording were carried out using a Multiclamp 700B amplifier (Molecular Devices, Sunnyvale, CA, USA), and Matlab (Mathworks, Natick, MA, USA) scripts calling NI-DAQmx library functions (National Instruments, Austin, TX, USA) to acquire and generate analog waveforms, using a National Instruments X-series DAQ interface. For current-clamp and voltage-clamp, the built-in series resistance compensation and capacitance cancellation circuitry of the Multiclamp were used. Pipettes (5–10 MΩ before sealing) were pulled from borosilicate glass capillaries (GC150F-7.5, Harvard Apparatus, Kent, UK), and, for single-channel recordings, coated with Sylgard (Dow Corning Europe, Belgium), and fire-polished. Signals were filtered at 6 kHz (−3 dB, 4-pole Bessel) and sampled at 20 kHz with 16-bit resolution. For conductance injection / dynamic-clamp (*Destexhe and Bal, 2009*) experiments, a hard real-time SM2 system (Cambridge Conductance, Cambridge, UK) was used, with low-latency AD and DA converters, and a digital signal processor (TMS C6713), running at a sample / update rate of > 50 kHz (<20 μs) (*Robinson, 2008*). Soma size (see Results) was used to select putative IS cells in experiments where spiking pattern was not assessed, e.g. cell-attached recordings.

## Statistics

All measurements are given as mean ± standard error of the mean (SEM), unless otherwise stated. To test for differences between two conditions, the two-sided Wilcoxon rank sum test (Matlab Statistics Toolbox function ranksum), equivalent to the Mann-Whitney U test, was used. n, the number of samples, and p, the probability of observing the two distributions under the null hypothesis that they have equal medians, are given in all cases, and p<0.05 is taken as the significance level.

## Spike analysis

Spike times were determined as the times of positive-going threshold crossings of the membrane potential at a threshold set at 10 mV below the peak of action potentials. Variability of the phase of spikes during sinusoidal stimulation (*Figure 8*) was characterized by a phase order parameter (*Pikovsky et al., 2001*) or synchrony of entrainment $S = \sqrt{\langle cos^2\phi \rangle + \langle sin^2\phi \rangle}$ (where $\phi$ is

the phase), which varied between 0 (phases distributed uniformly between 0 and $2\pi$) and 1 (phases all identical). Spike times within the first 250 ms of each response were omitted, to exclude initial adaptation from the analysis. The cross-correlation function of firing between gap-junction-coupled model neurons was calculated by binning times of occurrence of spikes in one cell relative to those in another cell (*Figure 7c*). The instantaneous firing rate was obtained by dividing the number of spikes in each time bin (0.5 ms) by the total simulation period and by the time bin. Synchrony of spikes between coupled model neurons (*Figure 7d*) was characterized as the fraction of spikes in one cell which occur within ± 10 ms of a spike in the other cell (obtained by integrating the cross-correlation function between −10 and +10 ms).

## Time series analysis

Recurrence plot (RP) analysis (*Eckmann et al., 1987*) was carried out using the Cross Recurrence Plot (CRP) Matlab toolbox (*Marwan et al., 2007*). Briefly, let the time series of sequential interspike intervals be indexed as $\{x_{1-m+1}, x_{1-m+2}, \ldots, x_1, x_2, \ldots, x_N\}$. For each trial, the initial transient in the first 450 ms was excluded, slow within-trial nonstationarity (<10% change in local average ISI) was removed by subtracting the least-square fit of the sequence to a second-order polynomial, and ISIs were normalized to zero mean, unit standard deviation. The state of the system at each interval $i$ can be represented by a vector of length $m$ of the immediately preceding intervals: $\vec{x}_i = [x_{i-m+1}, x_{i-m+2}, \ldots, x_i]$. The time series is said to be embedded with dimension $m$. The elements in the RP matrix are determined as follows

$$\mathbf{R}_{i,j} = \begin{cases} 1 : \vec{x}_i \approx \vec{x}_j, \\ 0 : \vec{x}_i \not\approx \vec{x}_j \end{cases} \quad i, j = 1, 2, \ldots, N,$$

where $N$ is the number of sequential states, and $\vec{x}_i \approx \vec{x}_j$ means equality within a distance (or error) of $\varepsilon$. Points of value 1 are plotted as black dots, 0 as white. Recurrence (for a given $\varepsilon$) is defined as the fraction of points in the RP which are 1, while determinism is the proportion of recurrent points which lie within diagonal lines of slope one and length greater than one. To measure distance between embedding points, we used a Euclidean norm (*Marwan et al., 2007*), and $\varepsilon$ was set at one standard deviation of the ISIs. The significance of both recurrence and determinism was measured by calculating the distribution of surrogate values obtained by randomly permuting the two time series in each cross-recurrence comparison, one thousand times, and a $z$-test to estimate the probability $p$ of obtaining the result by chance, with p<0.05 deemed significant.

## Fitting of voltage-clamped $g_{Kt}$ and dynamic conductance injection

A Hodgkin-Huxley-type model with one activation variable ($m$) and one inactivation variable ($h$) was fitted to step responses in voltage-clamp, such that $I_{Kt} = g_{Kt}(V - E_K)$. For dynamic conductance injection, three different parameter sets obtained from the experiments were used, differing slightly in the activation and inactivation kinetics, as follows.

### Models 1 and 2

$$\frac{dm}{dt} = \alpha_m(1-m) - \beta_m m \text{ and } \frac{dh}{dt} = \alpha_h(1-h) - \beta_h h, \text{ where}$$

### Model 1

$$\alpha_m(V) = \frac{0.0187*(V+52.5)}{1-\exp((52.5-V)/1.96)}, \qquad \beta_m(V) = 1.88*\exp((80.62-V)/74.36)$$
$$\alpha_h(V) = 0.0765*\exp((61.63-V)/9.16), \quad \beta_h(V) = \frac{0.0514}{1+\exp((83.86-V)/1.03)}$$

### Model 2

$$\alpha_m(V) = \frac{0.0175*(V+73.2)}{1-\exp((73.2-V)/5.59)}, \qquad \beta_m(V) = 1.47*\exp((68.6-V)/44.2)$$
$$\alpha_h(V) = 0.057*\exp((51.34-V)/29), \quad \beta_h(V) = \frac{0.054}{1+\exp((26.58-V)/23.72)}$$

Model 3

$$\frac{dm}{dt} = \frac{m_\infty - m}{\tau_m} \text{ and } \frac{dh}{dt} = \frac{h_\infty - h}{\tau_h}, \text{ with}$$

$$m_\infty(V) = \frac{1}{1+\exp((-30-V)/10)}, \qquad \tau_m(V) = 0.346\exp(-V/18.272) + 2.09$$

$$h_\infty(V) = \frac{1}{1+\exp(0.0878(V+55.1))}, \qquad \tau_h(V) = 2.1\exp(-V/21.2) + 4.627$$

All three models gave similar results, which are pooled together. To facilitate comparison across models and with a model-independent measure from experimental results, we characterize the amount of conductance measured in voltage-clamp, and injected with each model by the peak transient value reached at a potential of 0 mV, $g_{max0}$, rather than by $\bar{g}_{Kt}$.

## Fitting voltage-dependence of membrane potential noise

The onset of TTX-dependent voltage noise with depolarization (**Figure 3a**) was fitted by assuming that it was due only to non-inactivating (persistent) voltage-dependent sodium channels and inactivating $g_{Kt}$ channels, in a single passive cell compartment (i.e. without considering the effect of changes in the membrane potential on channel gating, including active amplification by the large Na conductance in the cell), whose passive conductance is $G$, and membrane time constant is $\tau_{cell}$. Let $\tau_m = 1/(\alpha_m + \beta_m)$ and $m_\infty = \alpha_m/(\alpha_m + \beta_m)$. Then by calculating the Lorentzian components of the single-channel noise expected from the Hodgkin-Huxley model, filtering with the membrane time constant and integrating over all frequencies (**Schneidman et al., 1998**), we obtain the following distribution of membrane potential variance, for NaP channels:

$$\sigma_V^2 = \frac{N^2 i^2 m_\infty^3}{G^2}\left[3m_\infty^2(1-m_\infty)\frac{\tau_m}{\tau_m + \tau_{cell}} + 3m_\infty(1-m_\infty)^2\frac{\tau_m/2}{\tau_m/2 + \tau_{cell}} + (1-m_\infty)^3\frac{\tau_m/3}{\tau_m/3 + \tau_{cell}}\right]$$

$N$ is the number of channels and $i$, the single channel current is given by $\gamma(V - E_{Na})$, where $\gamma$ is the single-channel conductance. For $g_{Kt}$ channels, $i = \gamma(V - E_K)$ and

$$\sigma_V^2 = \frac{N^2 i^2 m_{Kt,\infty} r_{Kt,\infty}}{G^2}\left[h_{Kt,\infty}\left(1-m_{Kt,\infty}\right)\frac{\tau_{mKt}}{\tau_{mKt} + \tau_{RC}} + m_{Kt,\infty}\left(1-h_{Kt,\infty}\right)\frac{\tau_{hKt}}{\tau_h + \tau_{RC}} + \left(1-m_{Kt,\infty}\right)\left(1-h_{Kt,\infty}\right)\frac{\tau_1}{\tau_1 + \tau_{RC}}\right]$$

where $\tau_1 = \tau_{mKt}\tau_{hKt}/(\tau_{mKt} + \tau_{hKt})$. The total membrane noise variance was taken as the sum of these two components.

## Model of irregular spiking

A reduced conductance-based model of IS neurons was implemented in Java (called from Matlab, see **Source code 1** in irregmodelcode.zip), based on a standard model of fast-spiking inhibitory interneurons (**Erisir et al., 1999**; **Gouwens et al., 2010**), to which was added a second compartment modeling dendritic membrane, a $g_{Kt}$ potassium conductance whose kinetics was obtained from fits to the voltage clamp data shown in **Figure 4**, and noise sources representing the effects of persistent sodium and $g_{Kt}$ channel openings. A somatic compartment, of capacitance $C$ = 8.04 pF and passive leak conductance $g_L$ = 4.1 nS, was connected with an intracellular resistance $R_i$ of 2 GΩ to a passive compartment representing the remote dendritic membrane, which had a capacitance $C_D$ of 80 pF and a leak conductance $g_D$ of 0.5 nS. Transient sodium (Na) and persistent sodium (NaP), Kv1 (K1), Kv3 and $g_{Kt}$ type potassium and static leak (L) conductances were inserted at the soma. The system of differential equations describing the model was as follows. The somatic voltage $V$ was determined by a Langevin equation containing noise terms $X$ for NaP and $g_{Kt}$ channel fluctuations:

$$C\frac{dV}{dt} = \left(\bar{g}_{Na}m^3 h + \bar{g}_{NaP}m^3\right)(E_{Na} - V) + \left(\bar{g}_{K1}n^4 + \bar{g}_{K3}p^2 + \bar{g}_{Kt}m_{Kt}h_{Kt}\right)(E_K - V) + g_L(E_L - V) + X_{NaP} + X_{Kt} + I_{stim}$$

The voltage of the passive dendritic compartment was determined by:

$$C_D\frac{dV_D}{dt} = \frac{(V - V_D)}{R_i} + g_D(E_L - V_D)$$

The kinetics of the gating variables of voltage-dependent channels were determined as follows (units of mV for voltage, ms$^{-1}$ for rates):

$$\frac{dx}{dt} = \alpha_x(V)(1-x) - \beta_x(V)x, \text{ for } x \in \{m,h,n,p\}, \text{ where}$$

$\alpha_m(V) = (3020 - 40V)/(\exp((-75.5 + V)/ - 13.5) - 1)$,

$\beta_m(V) = 1.2262/\exp(V/42.248)$,

$\alpha_h(V) = 0.0035/\exp(V/24.186)$,

$\beta_h(V) = -(0.8712 + 0.017V)/(\exp((51.25 + V)/-5.2) - 1)$,

$\alpha_n(V) = -(0.616 + 0.014V)/(\exp(-(44 + V)/2.3) - 1)$,

$\beta_n(V) = 0.0043/\exp((44 + V)/34)$,

$\alpha_p(V) = (95 - V)/(\exp(-(V - 95)/11.8) - 1)$,

$\beta_p(V) = 0.025/(V/22.222)$,

and $\frac{dm_{Kt}}{dt} = \frac{m_{Kt,\infty} - m_{Kt}}{\tau_{mKt}}$ and $\frac{dh_{Kt}}{dt} = \frac{h_{Kt,\infty} - h_{Kt}}{\tau_{hKt}}$, with

$m_{Kt,\infty}(V) = \frac{1}{1+\exp((-30-V)/10)}$ , $\tau_{mKt}(V) = 0.346\exp(-V/18.272) + 2.09$

$h_{Kt,\infty}(V) = \frac{1}{1+\exp(0.0878(V+55.1))}$, $\tau_{hKt}(V) = 2.1\exp(-V/21.2) + 4.627$

$\bar{g}_{Na}$=900 nS, $\bar{g}_{K1}$=1.8 nS, $\bar{g}_{Na}$=1800 nS, $g_L$=4.1 nS, $E_L$=−70 mV, $E_K$=−90 mV, $E_{Na}$=60 mV. Values of $\bar{g}_{Na}$, $\bar{g}_{K1}$ and $\bar{g}_{K3}$ are unchanged from those used for fast-spiking interneurons in *Erisir et al. (1999)* and *Gouwens et al. (2010)*, while $g_L$ was adjusted to give a resting input resistance similar to those measured in IS neurons. $\bar{g}_{Kt}$ was set to 7 nS or varied as described in the text (deterministic case), or 700 × 10 pS channels, or varied as described (stochastic case). $\bar{g}_{NaP}$ was set to 10 nS (deterministic) or 500 × 20 pS channels (stochastic case, see below).

The single persistent sodium channel current was given by $i = \gamma(E_{Na} - V)$ where $i$ is the single sodium channel current and $\gamma$ is the single channel conductance, set to 20 pS. Macroscopic persistent Na current was given by $I_{NaP} = \bar{I}_{NaP} + X$, where the deterministic mean current term was given by $\bar{I}_{NaP} = Nim^3$ in which $N$ is the number of persistent sodium channels (0 for the deterministic model, see text), and the noise term $X$ was updated at each time step by the exact update formula for an Ornstein-Uhlenbeck process (*Gillespie, 1996*):

$$X_{t+\Delta t} = X_t \exp(-\Delta t/\tau_o) + \xi\sqrt{\sigma_{NaP}^2(1 - \exp(-2\Delta t/\tau_o))}$$

where $\Delta t$ is the time step of integration, $\tau_o$ is the mean opening burst time of persistent Na channel openings, set at 1 ms, and $\xi$ is a normally-distributed (mean 0, variance 1) random number. The variance of $X$ changed dynamically, according to the mean level of persistent sodium current, as:

$$\sigma_{NaP}^2 = i\bar{I}_{NaP} - \bar{I}_{NaP}^2/N.$$

$g_{Kt}$ noise was modeled similarly, a single channel conductance of 10 pS, and a mean opening burst time of 10 ms, estimated from the recordings.

A fourth-order Runge-Kutta method (*Press et al., 2002*) was used to integrate deterministic variables, with a time step of 5 or 1 μs. The value of the noise term was updated in parallel, as described above, and interpolated linearly at the midpoint of full Runge-Kutta steps. This gave identical results to an Euler-Maruyama method (*Kloeden and Platen, 1992*), but with improved stability and efficiency.

## Acknowledgement

The authors would like to thank Paul Charlesworth (University of Cambridge) for providing primary cultures of dissociated cortical neurons.

## Additional information

### Funding

| Funder | Author |
| --- | --- |
| Biotechnology and Biological Sciences Research Council | Ole Paulsen Hugh PC Robinson |
| Coordenação de Aperfeiçoamento de Pessoal de Nível Superior | Philipe RF Mendonça |

| Cambridge Overseas Trust | Philipe RF Mendonça |
|---|---|

The funders had no role in study design, data collection and interpretation, or the decision to submit the work for publication.

## Author contributions

PRFM, HPCR, Conception and design, Acquisition of data, Analysis and interpretation of data, Drafting or revising the article; MV-C, Conception and design, Acquisition of data; FE, GS, Developed genetically-modified mouse; OP, Conception and design, Analysis and interpretation of data, Drafting or revising the article

## Author ORCIDs

Philipe RF Mendonça, http://orcid.org/0000-0003-1675-642X
Mariana Vargas-Caballero, http://orcid.org/0000-0003-2326-4001
Ole Paulsen, http://orcid.org/0000-0002-2258-5455
Hugh PC Robinson, http://orcid.org/0000-0002-5048-9954

## Ethics

Animal experimentation: Experimental procedures and animal use were in accordance with the animal care guidelines of the UK Animals (Scientific Procedures) Act 1986 under Home Office project license PPL80/2440 and personal licenses held by the authors. Caution was taken to minimize stress and the number of animals used in experiments.

## Additional files

**Supplementary files**
• Source code 1. Source code for model in *Figures 6* and *7*.

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
