## [Decision Letter]

Thank you for submitting your article "Stochastic and deterministic dynamics of intrinsically irregular firing in cortical inhibitory interneurons" for consideration by *eLife*. Your article has been favorably evaluated by Eve Marder (Senior editor) and three reviewers, one of whom, Frances K Skinner (Reviewer #1), is a member of our Board of Reviewing Editors. The following individuals involved in review of your submission have agreed to reveal their identity: Bruce P Bean (Reviewer #2) and Charles Wilson (Reviewer #3).

The reviewers have discussed the reviews with one another and the Reviewing Editor has drafted this decision to help you prepare a revised submission.

Summary:

This is a complex but most interesting paper on the dynamics of cells whose firing is intrinsically irregular. It is a thorough and detailed analysis made for a neuronal type showing how particular ion channels interact to control spiking patterns. The authors use a genetically altered mouse model to identify a specific type of cortical interneuron that is localized in layer 2 of the cortex and shows an unusual irregularly spiking phenotype, quite different from the better-known fast-spiking phenotype. The work is especially notable as the authors not only characterize some of the cell's biophysical characteristics, but they also create a model and use dynamic clamp to explore the irregularity mechanism and its potential role. These studies especially underlie the importance of determining biophysical specifics of particular cell types and not only doing fits to a generic set of conductances.

The analysis is effective at many levels: they describe a mechanism of irregular firing based on interaction of subthreshold Kv4 and persistent sodium currents. Ever since the classic papers by Connor and Stevens, previous analyses of the functional role of Kv4-mediated A-type potassium current have emphasized its role in enabling highly regular low-frequency firing, completely opposite to the critical role in enabling irregular firing that is discovered here. The authors use modeling to find that this mechanism confers on the neurons the ability to decrease synchronization at frequencies below 20 Hz, while permitting synchronization at higher (gamma band) frequencies, thus giving the neurons interesting network properties. The detailed analysis of how Kv4 transient A-type potassium current interacts with subthreshold persistent sodium current to produce irregularity of firing is very nicely done, comprising a detailed experimental characterization of the voltage-dependent and kinetic properties of the two conductance, use of that information in dynamic clamp experiments -which include a demonstration that the kinetics and voltage-dependence of the A-type current are well-enough characterized to allow subtraction of current, rarely attempted (or rigorously tested) in dynamic clamp experiments – and computer modeling that includes the stochastic behavior of the channels.

Essential revisions:

1) Confusing organization, incompleteness and lack of clarity in parts

There are a lot of pieces of this paper, and the connections between them are sometimes hard to follow. A suggestion is to include some statements at the beginning of the Results to guide the reader, and to consider some re-organization of the results. Statements below explain why the present organization is confusing. Other statements below point to where the paper needs to be made more clear and expanded with more information. In considering their re-organization, the authors also need to be clear about stochastic and deterministic aspects [specific statements for this are given in main point 2) below].

A) After describing the irregularity (Figure 2), without any warning, Figure 3 jumps to an entirely different topic, the origin of intrinsic noise. After reading Figure 2 one could be ready to believe that the irregularity might not be only noise. Why do we then immediately start looking for the origin of noise? The reader needs some help in following the thread of the logic. In any event, the result is that the noise is caused by the stochastic opening of ion channels, as in other neurons that fire more deterministically, and is no larger than in those neurons. At the end of this section we are invited to dismiss this noise as the cause of the irregular firing. Noise will appear again later, but it will not be required for irregular firing. Of course, at this point in the paper the reader does not know this.

B) Figure 4 introduces the presence of A currents in these neurons. This figure is not about irregularity, although the text begins by saying that blockers of calcium-dependent potassium channels did not alter irregularity (by the way, were blockers of Kv1 channels tried?). Data about Kv4 channels are then given, without any reference to their effect on irregularity. Blockers of Kv4 are used to study the current, but their effect on irregularity are not described. In fact, the effects of those blockers (4-AP or phrixotoxin) on irregularity of firing will never be presented. Why not? Certainly the authors have this information.

C) Figure 2 (and associated text) is confusing and could be more fully described. The description of the embedding dimension is unnecessarily confusing. From the beginning, when the authors are describing "two time series, A and B", it is unclear what these are. The time series are really series of spike times. Why not call them that? It would be much easier and much more interesting for neuroscientists. Are time series A and B spike times from two cells or only one? There is no indication. Are they the subsets of spike times from the same cell? The answers to these questions may seem obvious to the authors but they are not obvious to the readers, and they are essential for understanding the cross-recurrence plots. After they are understood, do the authors think that the reader will get some information from studying the cross-recurrence plots? If so, exactly what should they glean from them? Do they mean that some specific sequences of interspike intervals occur more often than expected by chance? If so, what kind of sequences are they? This figure is important, but as it is, it doesn't really add much to the paper from a reader's point of view. Later in the paper, a model will be presented that tends to fire in a particular set of preferred sequences. If this analysis gave us a first look at this, in the form of particular kinds of sequences that recur at higher than chance level, it would contribute strongly to the flow of the logic.

Also, the authors also say 'correlated' in the subsection “Recurrence of sequences of irregular interspike intervals”. At this point they were talking about recurrence and determinism, so this is confusing. What is intended?

D) In Figure 5, the effect of adding or subtracting Kv4-like currents at the soma are shown, and in this section they are shown to control irregularity. This is a key experimental part of the paper, and makes the case for the Kv4-like current as essential for irregularity. This part of the paper gets only an extremely brief description in the Results section despite its importance for the argument. And there is a little problem with it. Whereas adding the Kv4-like current was very effective at making the cell more irregular, deleting it was almost completely ineffective. Although the effect at -8.7 nS was apparently statistically significant, it is really a tiny change in regularity, and the cells' CV showed no systematic decrease over the range of negative conductances used. The test claims that the negative conductance produced a "striking regularization of firing", but this is not evident in the group measurements in Figure 5. There is apparently a regularization in the example shown in A and B. The change in CV is unconvincing in Figure 5. There are a little cluster of points at -0.1 nS/pF that are lower than the others, but in general, for points at conductances less than 0.2 nS/pF, it looks as though the removal of A current did not reduce CV. This is important for the argument, as the previous section did not show a change in regularity with Kv4 blockers. In the end, there is no strong evidence presented for the key point, which is that the natural level of Kv4 current is necessary for the irregularity.

For some values of *g_Kt_* – the larger ones – adding the stochastic element actually makes the CV smaller – though at smaller *g_Kt_* values, the CV increases. This brings attention to the experimental value of native *g_Kt_*, which is given as an average of 22 nS. This is much larger than the values that are "subtracted" using dynamic clamp (max of -8.7 nS) or the values in the model of Figure 7. Some comment on this seems called for. Perhaps this is why the effects of subtracting even 8.7 nS seem modest. Is there a reason that larger values were not subtracted? Perhaps because difficulties in exactly matching the kinetics would make it difficult to interpret?

E) After Figure 5, the paper shifts to a description of a model of the IS neuron, containing stochastic or deterministic versions of Nav and Kv4 currents, and also the Kv2 and Kv1 current. This transition was found to be confusing.

F) Also, the statements in the subsection “Voltage-dependent sodium channel openings are required for voltage fluctuations” to transition from the Na channels to K channels was confusing because of a mixing of noise, determinism and channel types. More specifics could be added regarding the CV values being compared in using this statement for example.

G) The authors use a 2-compartment model (passive dendrite second compartment) with particular capacitance and leak values (citing Erisir and Gouwens). Why did the authors use 2 compartments (unlike cited refs), and how were capacitance and leak values chosen (estimated from data in some way, or chosen to 'work')? Given the clear descriptions and rationales given for everything else in the paper (e.g., using 3 fitted models for voltage-clamped *g_Kt_* but that similar results were obtained), this aspect seemed lacking. Also, it could be more clearly stated that only the Kt current (putative Kv4) was directly fit from their data, and that the other ones were tuned (?), matched to data in some way? Since they are also showing their results with experimental work (dynamic clamp), it is ok if there were simply choices made for parameter values, but it would be helpful for the authors to be more explicit and expansive in describing their cellular model and rationale.

2) Stochastic and chaos aspects, clarity required

The Results and Discussion could benefit from a clearer summary of the relative importance of channel stochastics as opposed to chaotic deterministic behavior to the spiking irregularity.

A) It is noted that 'stochastic chaos' is mentioned in the cover letter, but this does not have a clear meaning/interpretation (separately, 'stochastic' and 'chaos' could be clear). However, 'chaos' or 'stochastic chaos' does not appear in the main manuscript. The authors should be clear and consistent in their terminology and interpretation and intention. For example, chaos could be shown to be present via Lyapunov exponent determination etc. Although doing this kind of analyses may not be critical here, being clear about terminology and methodology is important.

B) It is usually assumed that irregular firing arises because of irregularity in the synaptic barrage. Neurons have intrinsic noise, caused by the stochastic nature of ion channels, but the noise is relatively small and most cells fire in an almost deterministic way when the synaptic input is removed. Of course, chaotic firing has been demonstrated in realistic models of neurons (e.g. Canavier, Baxter, Clark and Byrne, J. Neurophysiol 1993, 69:2252-2257), but it is rare for authors to claim that irregularity is caused by chaos. This may be because even a small amount of noise can make the demonstration of chaos ambiguous.

C) One feature of the irregularly firing neurons are pauses, similar to those seen in fast spiking cells, and which are known in those cells to be caused by Kv1 channels. Adding the Kv1 channels to the model introduces pauses in firing that increase the CV. The model also shows subthreshold oscillations that also act to produce long ISIs, but these are of more variable duration because the oscillation occurs near threshold and is unstable. Every transit through the subthreshold oscillation provides an opportunity for another action potential, and this can produce repeating sequences in firing like those I guess are being detected in Figure 2 (although not sure because those sequences were never described). It is important that this phenomenon is independent of the presence of noise. The authors don't say so explicitly, but the model can clearly produce irregular firing via chaotic transitions between pauses, spiking, and unstable periodic orbits. In fact, the noise does not increase the irregularity, but rather decreases the range of parameters at which it is seen, as shown by comparing the extent of the red regions in Figure 7 and B. That figure also shows that the addition of noisy channels reduces the effect of Kv4 (gt) on CV.

D) The finding of Figure 8 is very interesting, but how much of the frequency specificity of entrainment is caused by irregularity per se, and how much to the intrinsic frequency of the unstable oscillations. Could the authors provide some comments/insight on this?

---

## [Author Response]

*Essential revisions:*

*1) Confusing organization, incompleteness and lack of clarity in parts*

*There are a lot of pieces of this paper, and the connections between them are sometimes hard to follow. A suggestion is to include some statements at the beginning of the Results to guide the reader, and to consider some re-organization of the results. Statements below explain why the present organization is confusing. Other statements below point to where the paper needs to be made more clear and expanded with more information. In considering their re-organization, the authors also need to be clear about stochastic and deterministic aspects [specific statements for this are given in main point 2) below].*

*A) After describing the irregularity (Figure 2), without any warning, Figure 3 jumps to an entirely different topic, the origin of intrinsic noise. After reading Figure 2 one could be ready to believe that the irregularity might not be only noise. Why do we then immediately start looking for the origin of noise? The reader needs some help in following the thread of the logic. In any event, the result is that the noise is caused by the stochastic opening of ion channels, as in other neurons that fire more deterministically, and is no larger than in those neurons.*

We have added a link sentence at the beginning of the section (“Voltage-dependent sodium channel openings are required for voltage fluctuations”), making it clearer that we are embarking on a systematic examination of biophysical mechanisms which could be relevant to the irregular firing, starting with the noise (and see answer to next point, below).

*At the end of this section we are invited to dismiss this noise as the cause of the irregular firing. Noise will appear again later, but it will not be required for irregular firing. Of course, at this point in the paper the reader does not know this.*

We certainly did not intend to give the impression of completely dismissing this noise, and the role of persistent Na channels – indeed our conclusion is eventually that they are an essential ingredient of the overall mechanism. To clarify this, we have added “also” in the final sentence of this section (“Voltage-dependent sodium channel openings are required for voltage fluctuations”): “another active mechanism might *also* be involved”. We have also revised the discussion of the model and overall mechanism to better bring out our view that irregular spike generation is best understood as an interaction of channel noise and the Hodgkin-Huxley type nonlinearly voltage-dependent currents near to firing threshold. (See the response to point 2C below).

*B) Figure 4 introduces the presence of A currents in these neurons. This figure is not about irregularity, although the text begins by saying that blockers of calcium-dependent potassium channels did not alter irregularity (by the way, were blockers of Kv1 channels tried?). Data about Kv4 channels are then given, without any reference to their effect on irregularity. Blockers of Kv4 are used to study the current, but their effect on irregularity are not described. In fact, the effects of those blockers (4-AP or phrixotoxin) on irregularity of firing will never be presented. Why not? Certainly the authors have this information.*

Kv1 channel blockers: we did try the effect of α-dendrotoxin on the transient outward current in voltage-clamp experiments in the presence of TTX (already described in the text, subsection “A fast-inactivating potassium current activates around threshold”, second paragraph), showing that it did not block the fast transient K current, but did not test its effect on irregularity.

Effects of Kv4 blockers on irregularity: We had some data which we did not discuss, and have since added some more. These data show that, indeed, phrixotoxin, and low-dose 4-AP both reduce irregularity of firing, in support of the identification of Kv4 as one of the controlling mechanisms. We have now added a sentence to this effect (subsection “Transient outward conductance determines spike irregularity”), referring to a new supplementary figure (new Figure 5—figure supplement 3), showing this.

*C) Figure 2 (and associated text) is confusing and could be more fully described. The description of the embedding dimension is unnecessarily confusing. From the beginning, when the authors are describing "two time series, A and B", it is unclear what these are. The time series are really series of spike times. Why not call them that? It would be much easier and much more interesting for neuroscientists. Are time series A and B spike times from two cells or only one? There is no indication. Are they the subsets of spike times from the same cell? The answers to these questions may seem obvious to the authors but they are not obvious to the readers, and they are essential for understanding the cross-recurrence plots. After they are understood, do the authors think that the reader will get some information from studying the cross-recurrence plots? If so, exactly what should they glean from them? Do they mean that some specific sequences of interspike intervals occur more often than expected by chance? If so, what kind of sequences are they? This figure is important, but as it is, it doesn't really add much to the paper from a reader's point of view. Later in the paper, a model will be presented that tends to fire in a particular set of preferred sequences. If this analysis gave us a first look at this, in the form of particular kinds of sequences that recur at higher than chance level, it would contribute strongly to the flow of the logic.*

*Also, the authors also say 'correlated' in the subsection “Recurrence of sequences of irregular interspike intervals”. At this point they were talking about recurrence and determinism, so this is confusing. What is intended?*

We have revised the text as suggested, re-expressing “time series” as interval sequences, and explaining what A and B concretely refer to (ISI sequences for two successive trials, as used later). Yes, it would be nice to understand what kinds of patterns are repeated. However, it would be misleading to suggest that we use this technique to actually identify and catalogue particular precise patterns, which is likely to be extremely difficult and inconclusive. Instead, we use this to test for statistical significance of the tendency of any complex patterns to recur. The neighbourhood size for detecting similar patterns is relatively large (one standard deviation of the ISIs, in 4-dimensional ISI embedding space). It is possible to use a much smaller neighbourhood, but the number of recurrent points then becomes much smaller, and it is correspondingly harder to test their significance.

We have made the following changes to address this point: We have changed Figure 2, first showing a recurrence plot in which distance is colour-coded (panel E), to bring out more clearly that it is based on Euclidean distance (as exemplified in panel D) and also, as suggested, selecting several examples of recurrent patterns of successive ISIs (new panel F) from various locations in Figure 2, to give a more concrete sense of what the analysis does. Then, we also provide a more complete demonstration of the basis of the significance testing, in a new supplementary figure (Figure 2—figure supplement 1), which shows one example of a cross-recurrence plot, a cross-recurrence plot of two corresponding shuffled surrogate time series of ISIs, and the corresponding histograms of recurrence values (panel C) and of determinism (panel D) from 1000 shuffled surrogates, compared to the actual measured values. This comparison is then the basis of the testing for significance, based on a z-test (as described in the Methods).

We hope that these changes will make the analysis clearer to readers unfamiliar with this approach, and they should give complete information about what was done, in conjunction with the cited references on the technique of recurrence plot quantification (Eckmann et al., 1997; Marwan et al., 2007).

*D) In Figure 5, the effect of adding or subtracting Kv4-like currents at the soma are shown, and in this section they are shown to control irregularity. This is a key experimental part of the paper, and makes the case for the Kv4-like current as essential for irregularity. This part of the paper gets only an extremely brief description in the Results section despite its importance for the argument. And there is a little problem with it. Whereas adding the Kv4-like current was very effective at making the cell more irregular, deleting it was almost completely ineffective. Although the effect at -8.7 nS was apparently statistically significant, it is really a tiny change in regularity, and the cells' CV showed no systematic decrease over the range of negative conductances used. The test claims that the negative conductance produced a "striking regularization of firing", but this is not evident in the group measurements in Figure 5. There is apparently a regularization in the example shown in A and B. The change in CV is unconvincing in Figure 5. There are a little cluster of points at -0.1 nS/pF that are lower than the others, but in general, for points at conductances less than 0.2 nS/pF, it looks as though the removal of A current did not reduce CV. This is important for the argument, as the previous section did not show a change in regularity with Kv4 blockers. In the end, there is no strong evidence presented for the key point, which is that the natural level of Kv4 current is necessary for the irregularity.*

*For some values of g_Kt_ – the larger ones – adding the stochastic element actually makes the CV smaller – though at smaller g_Kt_ values, the CV increases. This brings attention to the experimental value of native g_Kt_, which is given as an average of 22 nS. This is much larger than the values that are "subtracted" using dynamic clamp (max of -8.7 nS) or the values in the model of Figure 7. Some comment on this seems called for. Perhaps this is why the effects of subtracting even 8.7 nS seem modest. Is there a reason that larger values were not subtracted? Perhaps because difficulties in exactly matching the kinetics would make it difficult to interpret?*

We appreciate these comments, and agree that increased regularity by cancellation of native *g_Kt_* is an important point to demonstrate. However, we do not agree that this is a minimal or unclear effect. Firstly, the measurements as shown in original Figure 5, for -8.7 nS injection, gave a clear and significant reduction. The data for the -4.35 nS histogram bar were not significant, but there was a relatively low n for that set of measurements (n = 7). In the original Figure 5, we used data from only one of the three gKtconductance definitions (model 3). In the new Figure 5, we have incorporated data from the two other models as well (n=42 cells), and instead tested the lumped negative and positive conductance injections against control. Both results are significant, with *p*<1.6 X 10^-8^ for negative injections, and *p*<9.8 X 10^-16^ for positive injections. The basis for this can be seen in a clearly shifted set of points for the negative injection case (red points). In a new supplementary Figure 5—figure supplement 2, we now show several examples of the changes in spike train pattern induced by both negative and positive *g_Kt_* injection. In Figure 5, plotting the relative change in CV(ISI) versus normalised conductance injected, the trend is clear across both positive and negative conductance injections, and we have now added a linear regression fit to emphasise this. The apparent cluster of points at ≈-0.1 nS/pF is real – all of those cells indeed showed a relatively big reduction in CV(ISI) for what turned out (after normalisation) to be a relatively modest negative injection. Note that there is less scope to reduce CV(ISI) experimentally by negative conductance injection than to increase it by positive conductance injection, because we are limited in the amount of negative conductance that we can inject without creating instability. Also, as irregularity is reduced by cancelling *g_Kt_*, other sources of variability may start to dominate. Despite this, we were able to cancel almost half of the native *g_Kt_* (8.7 nS of 22 nS), and this had a significant effect in reducing CV(ISI). Therefore, we feel it is safe to conclude that the native *g_Kt_* has a prominent role in producing the experimental level of irregularity.

*E) After Figure 5, the paper shifts to a description of a model of the IS neuron, containing stochastic or deterministic versions of Nav and Kv4 currents, and also the Kv2 and Kv1 current. This transition was found to be confusing.*

In the first paragraph of the subsection “Mechanisms of firing variability in a simple model of IS neurons”, we have now introduced the shift to modelling more clearly, and we hope with more evident rationale, including a statement that we will study the key *g_Kt_* and NaP conductances either as stochastic or deterministic elements.

*F) Also, the statements in the subsection “Voltage-dependent sodium channel openings are required for voltage fluctuations” to transition from the Na channels to K channels was confusing because of a mixing of noise, determinism and channel types. More specifics could be added regarding the CV values being compared in using this statement for example.*

We looked in the literature for relevant CV(ISI) measurements on the entorhinal cortex stellate cells studied by White et al., and could not find any specific values at comparable firing frequencies (although CV(ISI) is measured during lower frequency firing by Burton et al., 2008, J Neurophysiol 100:3144). However, it is clear from recordings such as Figure 7 (top trace) of Alonso and Klink (1993), which we have now referred to, that the variability of firing at around 10 Hz is considerably less than in IS cells. Thus we have now simply described the comparison as “… but without appearing to produce the high level of firing irregularity in IS cells”. We believe that this is the issue referred to by the Reviewers.

*G) The authors use a 2-compartment model (passive dendrite second compartment) with particular capacitance and leak values (citing Erisir and Gouwens). Why did the authors use 2 compartments (unlike cited refs), and how were capacitance and leak values chosen (estimated from data in some way, or chosen to 'work')? Given the clear descriptions and rationales given for everything else in the paper (e.g., using 3 fitted models for voltage-clamped g_Kt_ but that similar results were obtained), this aspect seemed lacking. Also, it could be more clearly stated that only the Kt current (putative Kv4) was directly fit from their data, and that the other ones were tuned (?), matched to data in some way? Since they are also showing their results with experimental work (dynamic clamp), it is ok if there were simply choices made for parameter values, but it would be helpful for the authors to be more explicit and expansive in describing their cellular model and rationale.*

The model was based on the Erisir et al., 1999, model of FS neurons, probably the most widely-studied biophysical model of interneuron excitability, with the addition of the *g_Kt_*/Kv4 current, whose characteristics were based on those measured. As we have now made explicit in the Methods, parameters and conductance values for Na, Kv1 and Kv3 were unchanged from the Erisir et al. model, as they gave realistic action potential waveforms and levels of excitability, while the leak conductance was adjusted to give a typical input resistance for IS neurons. As for the inclusion of the dendritic compartment, we have now added an explanation in the first paragraph of the subsection “Mechanisms of firing variability in a simple model of IS neurons” in the Results section that this was done in order to capture some of the electrical consequences of the fact that the neurons have an extended morphology. It also means that the model (whose full code is provided as supplementary material) has parameters that can be used to modify and study this spatial aspect (although we did not do so), and it can also be readily adapted to study the effect of synaptic input in the dendritic compartment.

*2) Stochastic and chaos aspects, clarity required*

*The Results and Discussion could benefit from a clearer summary of the relative importance of channel stochastics as opposed to chaotic deterministic behavior to the spiking irregularity.*

*A) It is noted that 'stochastic chaos' is mentioned in the cover letter, but this does not have a clear meaning/interpretation (separately, 'stochastic' and 'chaos' could be clear). However, 'chaos' or 'stochastic chaos' does not appear in the main manuscript. The authors should be clear and consistent in their terminology and interpretation and intention. For example, chaos could be shown to be present via Lyapunov exponent determination etc. Although doing this kind of analyses may not be critical here, being clear about terminology and methodology is important.*

*B) It is usually assumed that irregular firing arises because of irregularity in the synaptic barrage. Neurons have intrinsic noise, caused by the stochastic nature of ion channels, but the noise is relatively small and most cells fire in an almost deterministic way when the synaptic input is removed. Of course, chaotic firing has been demonstrated in realistic models of neurons (e.g. Canavier, Baxter, Clark and Byrne, J. Neurophysiol 1993, 69:2252-2257), but it is rare for authors to claim that irregularity is caused by chaos. This may be because even a small amount of noise can make the demonstration of chaos ambiguous.*

*C) One feature of the irregularly firing neurons are pauses, similar to those seen in fast spiking cells, and which are known in those cells to be caused by Kv1 channels. Adding the Kv1 channels to the model introduces pauses in firing that increase the CV. The model also shows subthreshold oscillations that also act to produce long ISIs, but these are of more variable duration because the oscillation occurs near threshold and is unstable. Every transit through the subthreshold oscillation provides an opportunity for another action potential, and this can produce repeating sequences in firing like those I guess are being detected in Figure 2 (although not sure because those sequences were never described). It is important that this phenomenon is independent of the presence of noise. The authors don't say so explicitly, but the model can clearly produce irregular firing via chaotic transitions between pauses, spiking, and unstable periodic orbits. In fact, the noise does not increase the irregularity, but rather decreases the range of parameters at which it is seen, as shown by comparing the extent of the red regions in Figure 7 and B. That figure also shows that the addition of noisy channels reduces the effect of Kv4 (gt) on CV.*

We have reorganised and rewritten parts of the Discussion to make the points raised in a clearer way. Our view is that the noise and the nonlinearity are both essential in determining the actual level and characteristics of irregularity. We certainly agree that the noise-free deterministic model actually leads to higher irregularity in some parameter regions of putatively chaotic behaviour. We now state that this behaviour of the fully deterministic model resembles chaos, but that a rigorous demonstration of this by estimating Lyapunov exponents is beyond the scope of this study (Discussion, sixth paragraph). We now highlight, in the Results and Discussion, the fact that this CV(ISI) is unnaturally high (≈1) and that the addition of the noise reduces it to the experimental value (≈0.3), while still clearly preserving the existence of the high CV region induced by *g_Kt_*. We conclude in the Discussion that “Thus, we believe that the interaction of both elements, the nonlinear deterministic Hodgkin-Huxley equations and the single-channel dynamical noise, is needed for an adequate description of irregular spiking.”.

*D) The finding of Figure 8 is very interesting, but how much of the frequency specificity of entrainment is caused by irregularity per se, and how much to the intrinsic frequency of the unstable oscillations. Could the authors provide some comments/insight on this?*

Yes, thank you for this suggestion. We have now added a sentence in the description of Figure 8 in the Results subsection “Synchronization to oscillating input and the function of IS neuron”, stating that while the rejection of synchrony at lower frequencies surely results from the irregular dynamics, the synchrony at higher frequencies could be assisted by resonance with the noise-masked subthreshold oscillation, whose frequency in the deterministic model was determined as 28 Hz.